# Discovering New QTNs and Candidate Genes Associated with Rice-Grain-Related Traits within a Collection of Northeast Core Set and Rice Landraces

**DOI:** 10.3390/plants13121707

**Published:** 2024-06-19

**Authors:** Debjani Roy Choudhury, Avantika Maurya, Nagendra Kumar Singh, Gyanendra Prata Singh, Rakesh Singh

**Affiliations:** 1Division of Genomic Resources, ICAR—National Bureau of Plant Genetic Resources, New Delhi 110012, India; roydebj@gmail.com (D.R.C.); avantika.maurya@gmail.com (A.M.); 2ICAR—National Institute for Plant Biotechnology, New Delhi 110012, India; nksingh4@gmail.com; 3ICAR—National Bureau of Plant Genetic Resources, New Delhi 110012, India; gp.singh@icar.gov.in

**Keywords:** multi-locus GWAS, quantitative trait nucleotides, core, enrichment analysis, candidate genes, rice

## Abstract

Grain-related traits are pivotal in rice cultivation, influencing yield and consumer preference. The complex inheritance of these traits, involving multiple alleles contributing to their expression, poses challenges in breeding. To address these challenges, a multi-locus genome-wide association study (ML-GWAS) utilizing 35,286 high-quality single-nucleotide polymorphisms (SNPs) was conducted. Our study utilized an association panel comprising 483 rice genotypes sourced from a northeast core set and a landraces set collected from various regions in India. Forty quantitative trait nucleotides (QTNs) were identified, associated with four grain-related traits: grain length (GL), grain width (GW), grain aroma (Aro), and length–width ratio (LWR). Notably, 16 QTNs were simultaneously identified using two ML-GWAS methods, distributed across multiple chromosomes. Nearly 258 genes were found near the 16 significant QTNs. Gene annotation study revealed that sixty of these genes exhibited elevated expression levels in specific tissues and were implicated in pathways influencing grain quality. Gene ontology (GO), trait ontology (TO), and enrichment analysis pinpointed 60 candidate genes (CGs) enriched in relevant GO terms. Among them, *LOC_Os05g06470*, *LOC_Os06g06080*, *LOC_Os08g43470*, and *LOC_Os03g53110* were confirmed as key contributors to GL, GW, Aro, and LWR. Insights from QTNs and CGs illuminate rice trait regulation and genetic connections, offering potential targets for future studies.

## 1. Introduction

Grain-quality-related traits like grain length, grain width, and aroma are pivotal agronomic traits in rice and have evolved through natural selection during domestication. Initially, the improvement in grain size was an unintentional adaptation to rice cultivation, favoring seeds capable of thriving in deeper soil [1]. Interestingly, this trait was subsequently subjected to deliberate selection and breeding due to its direct impact on rice grain quality and overall yield [2,3]. The contemporary landscape of rice varieties now displays a diverse spectrum of grain size characteristics, which predominantly determine their yield. The dimensions of rice grains encompass grain length (GL), grain width (GW), and grain thickness (GT), as outlined in earlier studies [4,5]. These characteristics are intricately linked to grain weight, a significant factor influencing overall yield, in conjunction with the quantity of panicles per plant and the number of grains per panicle. The intricate nature of rice grain traits has undergone thorough investigation, revealing a complex inheritance system regulated by numerous genes that exert gradual and modest influences [6]. Researchers have employed both forward and reverse genetics to gain a deeper understanding of the genetic basis of these traits [7,8]. Multiple mapping experiments have been conducted using populations resulting from crosses between two parents to reveal the specific genomic areas accountable for grain size. Quantitative trait loci (QTL) related to LWR in indica rice have been detected previously [6,9]. Over 500 QTLs associated with grain-size-related traits, including grain length, grain width, and thousand-grain weight (TGW), have been mapped across all rice chromosomes [10]. However, few of these QTLs have undergone fine-mapping. QTL *gw8.1* [11], *gw3.1* [12], *qGL3-a* [13], *gw5* [14], *gw9.1* [15], *GW1-1*, *GW1-2* [16], *GW3*, *GW6*, *qSS7* [17,18] *qGS7* [19], *qGRL1* [20], *qGL7* [21], *qGL4b* [22], and *GS2* [23] have been fine mapped. *GS3* [24], *TGW6* [25], *GW2* [26], *qGL3/qTGW3* [27], and *LARGE8* [28] are negative regulators of grain size. On the other hand, *GS2* [29,30], *BG1* [31], *GS9* [32], and *qLGY3/OsLG3b* [33], *GW5/qSW5* [34,35], *GS5* [36], and *WTG1* [37,38] are positive regulators of grain size. It is worth noting that these cloned genes influence grain size by modifying cell proliferation and/or cell expansion, impacting the number of cells either in the latitudinal or longitudinal directions. Apart from GL and GW, aroma also forms a very important grain characteristic, with high commercial value. Rice quality can be assessed based on its commercial value, nutritional value, or its sensory attributes [39]. Aroma is a highly esteemed quality trait in rice, contributing to its elevated market value both nationally and internationally [40,41]. The distinctive features of aromatic rice varieties are attributed to volatile organic compounds [42]. Aroma, therefore, is a blend of more than 500 volatile chemical compounds that collectively shape the fragrance of rice [39]. Notably, among these biochemical compounds, 2-acetyl-1-pyrroline (2AP) has been identified as the most significant contributor to rice flavor [43]. The concentration of 2AP is particularly high in the leaves during the early growth stages, reaching its peak at the booting stage and subsequently diminishing in the leaves during the reproductive stage [41]. Importantly, 2AP is transported from the leaves and stem sheath and accumulates in the grains of aromatic rice varieties, hence the characteristic fragrance observed in aromatic rice [43]. Genetic studies have unveiled various genetic factors influencing rice aroma. While some point out to a single recessive gene [44,45], others suggest a dominant gene or multiple genes regulating this particular trait. Among these findings, three QTLs have been linked to aroma in rice. The most noteworthy QTL, *qaro8.1*, situated on chromosome 8 [44], holds particular significance in this regard. Additionally, *qaro3-1*, another notable QTL associated with aroma, has been pinpointed on chromosome 3 [46]. Moreover, specific alleles of the badh1 gene, found within the *qaro4.1* QTL on chromosome 4, were found to be correlated with aroma in various rice varieties [44].

Recently, ML-GWAS methods have been very effective and useful for association analysis. Methods include FASTmrEMMA, ISIS EM-BLASSO, mrMLM, FASTmrMLM, and pLARmEB [47,48]. These have been used to decipher novel QTNs in many crops. In wheat, five ML-GWAS models uncovered new QTLs for yield-related traits in 272 local Chinese wheat landraces using 172,711 SNPs [49]. A ML-GWAS of 144 inbred maize lines genotyped with 43,427 SNPs identified a multitude of significant QTNs and 40 candidate genes associated with embryonic callus regeneration [50]. ML-GWAS has been utilized in rice, identifying 74 significant QTN hotspots linked to five yield-related traits [51]. The strength and variety of ML-GWAS models offer promising potential for crop improvement by uncovering new loci and candidate genes associated with yield traits. Advances in rice functional genomics have allowed for the characterization of genes that regulate grain size, both positively and negatively [32]. However, only a limited number of these QTLs have practical applications in plant breeding. This is because the effects estimated for identified QTLs are often specific to the parental lines studied, as most genetic mapping research depends on conventional linkage mapping with populations from bi-parental crosses. Therefore, capturing genetic variation for quantitative traits like grain size requires an approach that leverages historical recombination events through linkage disequilibrium [52]. Genome-wide association study is a prominent quantitative genomics tool that has gained prominence in the mapping of quantitative traits with high-density genotyping platforms. Therefore, performing association studies through high-density SNP chips has enabled the selection of potential candidate genes for GWAS. There have been substantial efforts on breeding lines to decipher new QTLs for target traits. However, limited efforts have been made with germplasms to characterize genomic regions associated with target traits. In this study, genome-wide association mapping was conducted with a statistically strong and diverse association panel of the northeast core set (190 accessions) [53] and a set of rice landraces (293 accessions) [54] taken together to identify significant marker–trait associations for grain-related traits (GL, GW, Aro, and LWR). This study offers crucial insights for the continued exploration of elite genes within the northeast core and landrace sets for utilization in rice breeding. The findings of this study may be useful in further elucidating the genetic basis of rice grain size for improving grain characteristics in rice. These selected genes offer a window into delving deeper into the genetic framework of grain quality characteristics, thereby contributing to enhancing genetic improvements. 

## 2. Results

### 2.1. Trait Correlations and Variance 

Four grain-related traits, GL, GW, Aro, and LWR, were investigated in the selected set of 483 rice panel. The list of 483 rice panels has been included in Appendix A. The diverse set of rice accessions showed a wide range of values for all the grain-related traits. Aroma analysis showed a smaller number of samples being aromatic. A wide range of values for all the grain-related traits was recorded, and their descriptive statistics are represented in Table 1. The mean values of GL, GW, and LWR were 5.6, 2.4, and 2.3, respectively. The skewness of the population showed negative skewness for GL and GW traits, whereas LWR and aroma showed positive skewness. Kurtosis was less than 3 for GL, GW, and LWR, indicating a platykurtic distribution of phenotypes in the population, which means traits are governed by the large number of genes [6]. 

The correlation analysis was performed to understand the linear relationship between grain traits. The correlation was positive, significant, and strong between GL and GW (0.31) and between GL and LWR (0.42). Negative, strong, and significant correlation was observed with GW and LWR (−0.71), which means they are inversely proportional. The rest of the correlation values did not turn out to be significant. The results suggest a close relationship between the mentioned traits and indicate their possible contribution to enhancing the genetic improvement of rice (Figure 1).

### 2.2. Genetic Structure and Linkage Disequilibrium Analysis

Before conducting GWAS, the genetic architecture of the 483-rice panel was assessed using principal component analysis (PCA), kinship analysis, a neighbor-joining (NJ) tree, and population structure analysis. The SNP chip DNA marker density across all 12 rice chromosomes is illustrated in Figure 2a. Additionally, the pairwise linkage disequilibrium (LD) between markers, represented by the average r^2^ values of bins plotted against the physical distance between markers, produced a near-flat curve, indicating relatively low LD decay in the population. The maximum r^2^ value (r^max^) obtained was ~0.48, and the r^max^ was reduced to half at a distance, i.e., 133.46 kbp (Figure 2b). Therefore, 133.46 was calculated to be the LD distance.

### 2.3. PCA, Kinship and Relatedness Study

According to principal component analysis, there were three subpopulations in the selected rice panel (Figure 3a). PC1 explained 15.5% of the variance and PC2 explained 12.5% of the variance. The kinship matrix was computed to assess genetic relationships within the population. The coefficient of relatedness spanned from −2.0 to 2.0, and a kinship heat map was generated to illustrate these relationships. Notably, the upper right corner exhibited a relatively close relationship, while the rest of the accessions displayed lower coefficients of relatedness showing them to be unique and different (Figure 3b). Further, NJ tree also unveiled three distinct clusters, from the genetic distances derived from SNP variations among the selected rice accessions. Cluster 1 and 2 comprised 26 accessions with the majority of the accessions from Arunachal Pradesh, Nagaland, Tripura, Mizoram, Meghalaya, Manipur, and Assam. Cluster 3 comprised a maximum number of samples (431), with the majority of the Uttarakhand samples being grouped. The accessions from the rest of the states showed mixing. This observation suggests that the population utilized in our study confirms a natural population, with only a few instances of close relatedness among certain accessions. Concordantly, STRUCTURE also showed three subpopulations within the 483-panel based on the distribution of the 35,286 SNPs across 12 rice chromosomes. Appendix A shows the structure bar plot with three subpopulations as revealed by Structure Harvester. The genotypes exhibiting ≥80% likelihood was designated pure, whereas others were categorized as admixtures. The three subpopulations were accommodating pure and admixed individuals. Subpopulation 1 had 74 pure and 42 admixed individuals, subpopulation 2 had 43 pure and 21 admixed individuals, and subpopulation 3 had 317 pure and 28 admixed individuals. Overall PCA, kinship, and relatedness studies showed the accessions to be highly diverse.

### 2.4. GWAS and CGs Mining for Grain-Related Traits

A total of 40 QTNs were detected for four grain-related traits: GL, GW, grain aroma, and LWR. QTNs with LOD scores > 3.0 were considered significant trait-related QTNs. Most of the QTNs were identified with at least two of the five ML-GWAS methods, namely, mrMLM, FASTmrEMMA, FASTmrMLM, pLARmEB, and ISIS EM-BLASSO, utilized in the study. Nineteen of the QTNs detected overlapped with previously reported QTL/genes, which shows the consistency of the study. The number of QTNs detected varied with various methods. The Manhattan and quantile–quantile plots of all four traits presented in Figure 4 indicate that false associations were controlled, and the SNPs detected by ML-GWAS methods were true associations. A positively skewed QQ plot was observed in the case of aroma, which means that the observed *p*-values are more extreme than expected under the null hypothesis, which could indicate the presence of true associations between the genetic variant and the trait of interest. Eight QTNs were detected to be associated with GL on chromosomes 1, 2, 4, 5, 8, and 12, with LOD scores ranging from 3.04 to 5.72. There were nine QTNs identified for GW with LOD scores ranging from 3.14 to 5.29. For aroma, 12 QTNs were identified on chromosomes 1, 3, 5, 8, 10, and 11, with LOD scores ranging from 3.04 to 5.88. A maximum number of QTNs was detected on chromosome 8. For LWR, 11 QTNs were identified with LOD scores ranging from 3.0 to 6.6. Of these 40 QTNs, 38 QTNs were observed to be in the vicinity of annotated genes, and 16 QTNs were identified simultaneously by two or more ML-GWAS methods (Table 2; 16 QTNs have been marked in bold). Further, all 16 genomic loci associated with grain quality traits were searched for their annotation in Rice Assembly version 7. Probable CGs were searched in the 130-kbp genomic region of each 16 commonly annotated QTNs. For 16 QTNs, 258 genes were detected closer to significant QTNs. A Venn diagram was prepared showing the number of overlapping QTNs by various methods (Appendix A).

### 2.5. Superior Allele Distribution in Northeast Core and Rice Landraces

All 40 QTNs were studied for their superior allele and inferior allele in the rice landrace and northeast core to identify better-suiting genotypes and to see their distribution in the two sets. For GL among the eight QTNs reported, among them, SNP AX-95915857, AX-95947399, AX-95926370, AX-95925933, and AX-95930845 showed the presence of superior alleles to 80% in the NE core set, whereas SNP AX-95936094 was present in 30% of the accessions; therefore, this QTN needs attention for improvement of GL. For GW SNPs, AX-95946823, AX-95952472, AX-95936134 (Figure 5a), AX-95927762, and AX-95930391 showed their presence in less than 30% of the accessions in the northeast core, indicating that these SNPs should be paid more attention for improvement of the GW trait. Ten SNPs (AX-95918031, AX-95941053, AX-95921840, AX-95954140, AX-95926338, AX-95930157, AX-95937862, AX-95930471, AX-95960288, and AX-95932719) showed the presence of superior alleles in fewer than 30% of the accessions; hence, more attention needs to be paid to these loci for aroma improvement. LWR QTNs, SNP AX-95918592, AX-95920263, and AX-95947983 showed low percentages of superior alleles in the northeast core, suggesting that more study is needed for these QTNs for improvement of the LWR in the collection.

For the landraces set (Figure 5b), SNPs AX-95915857, AX-95947399, AX-95926370, AX-95930845, AX-95930845, AX-95963880, and AX-95964306 are present in more than 60% of the accessions, which can account for its rice diversity; however, AX-95936094 and AX-95925933 are present in less than 30% of the rice landrace collection, which means that these SNPs need more study to improve the GL trait in the collection. For the GW trait, superior SNPs (AX-95946823, AX-95955084, AX-95927762, and AX-95933306) were in high percentages, whereas AX-95952472, AX-95936134, AX-95955169, AX-95930391, and AX-95932902 were in very low percentages in the rice landrace collection, and these need improvement to improve the GW trait. In the case of aroma, most of the superior alleles (AX-95941053, AX-95918031, AX-95930157, AX95930471, and AX-95932719) were present to less than 50% of the accessions in the rice landrace collection, which calls for improvement to improve the quality of aromatic rice in this collection. With respect to the LWR trait, SNPs AX-95948231 and AX-95952612 were present in more than 60% of the accessions, which counts for good diversity in LWR in the rice landrace collection; however, SNPs AX-95918592, AX-95920263, AX-95947983, AX-95924497, and AX-95957972 need more study to improve LWR in the collection. The above distribution suggests that both the sets are rich in alleles and provides valuable insights into the allelic distribution.

### 2.6. Gene Functional Enrichment Analysis

To delve deeper into the various loci linked to the desired trait, we conducted an enrichment analysis using PlantGSAD. Furthermore, we predicted the potential functions of 60 candidate genes (CGs) and categorized them according to their respective molecular functions, biological processes, and cellular components (Table 3). Additionally, we utilized REVIGO [55] (http://revigo.irb.hr/) to analyze and generate non-repetitive gene ontology (GO) terms, which were then illustrated via a scatterplot based on the frequency and *p*-value of the GO terms (Figure 6). The bubble size and the value of the scatterplot depict the term significance. In total, 20, 17, and 29 terms were significantly enriched in the biological processes, molecular function, and cellular components, respectively. The GO term “regulation of biological quality” (GO:0065008, FDR = 1.30 × 10^−11^) was significantly enriched, indicating that the overlapping genes modulate a qualitative or quantitative trait of a biological quality (Appendix A). The overlapping genes include *LOC_Os08g43470* (qAro-8-4), which was found to be significantly associated with trait aroma, and *LOC_Os05g06460* and *LOC_Os05g06430*, which are neighboring genes to QTN found for GL (qGL-5-1). CGs were enriched in other biological processes, such as cellular homeostasis, biological regulation, maintenance of location in the cell, positive regulation of gene expression, and cellular metabolic processes, indicating their role in cell development processes affecting grain quality trait. The GO:0051173 (positive regulation of nitrogen compound metabolic process) term overlapped with *LOC_Os10g10990*, suggesting its role in influencing the aroma, as shown in a previous study where grain 2-acetyl-1-pyrroline (2-AP) biosynthesis in the presence of nitrogen application at the booting stage was seen to be enhanced [56]. Furthermore, molecular and cellular components were annotated and detected to be enriched in the 60 CGs. GO:0004148 (dihydrolipoyl dehydrogenase activity; FDR = 1.78 × 10^−9^), a multifunctional oxidoreductase, and GO:0003924 (GTPase activity; FDR = 1.42 × 10^−3^), involved in the G-protein signaling pathway that governs cell expansion and proliferation, were also determined. Cellular processes include the proteasome core complex, the latherin coat of the coated pit, the trans-Golgi network transport vesicle, the intracellular organelle, and the cell itself. *LOC_Os03g53110* and *LOC_Os06g06030* were annotated as constituent parts of a cell (GO:0044464). GO: 0005839 (Proteasome core complex; FDR = 6.24 × 10^−17^) has a role in peptide cleavage at C-terminal of hydrophobic, basic, and acidic residues. This multi-functional enzyme complex plays a role in the ubiquitin–proteasome pathway, regulating cell cycle progression. Hence, these findings demonstrate the impact of these CGs on rice grain size traits and grain aroma.

For further analysis of candidate genes, we applied the PlantGSAD platform, focusing on the TO category, chromatin states, and pathways (Appendix A), to predict the potential functions of CGs related to agronomic traits. As shown in Figure 7, various developments related to TO terms were significantly enriched, suggesting the putative role of CGs in the regulation of grain quality. TO:0000975 (grain width), TO:0000734 (grain length), and TO:0000587 (endosperm quality) were identified to be significantly enriched (Figure 7). *LOC_Os05g06480* (inorganic H+ pyrophosphatase), *LOC_Os06g06050* (F-box/LRR-repeat protein), and *LOC_Os06g06090* (mitogen-activated protein kinase 6) were found to be associated with the enriched TO terms. Inorganic H+ pyrophosphatase has shown its role in cell size and seed development [57]. It has been demonstrated earlier that overexpression of F-box proteins reduced the levels of ethylene and has promoted hull cell expansion and increased grain size [58]. Also, mitogen-activated protein kinases (OsMAPK6), mainly located in the nucleus and cytoplasm, are ubiquitously distributed in various organs, predominately in spikelet and spikelet hulls, and have a consistent role in rice grain size enhancement [31]. Hence, SNPs found to enrich TO terms can form causal variants or associated variants and establish a comprehensive collection of standardized trait vocabularies and descriptions, particularly for uncharacterized traits in crops.

The identification and understanding of plant chromatin states could provide valuable insights into the locations and roles of regulatory regions and genes, particularly in response to developmental cues and environmental stimuli. We studied chromatin states of the candidate loci using PlantGSAD and plant chromatin state database [59]. CGs were analyzed for their association with epigenetic markers of gene activation (H3K36me1, H3K36me2, H3K36me3, and H3K4me3) (Appendix A). Co-enrichment of H3K36me2 and H3K36me3 (combining effect) should be correlated with higher transcriptional activity [60]. Histone acetylation was also prominently associated with CGs. Previous studies have demonstrated that histone acetylation increases grain size by positively regulating hull cell proliferation [58]. 

For pathway analysis, we opted for MapMan to discern the pathways and processes linked to CGs, given its specificity in covering plant-specific pathways. CGs were identified as overlapping with protein degradation, protein synthesis, and protein targeting secretory pathways (Figure 8). Protein ubiquitination is involved in grain development via the cascade pathway involving ubiquitin-activation (E1), ubiquitin-conjugation (E2), and ubiquitin ligation (E3) enzymes, regulating proteasomal degradation, protein stability, and localization [61]. Therefore, CGs are found to be associated with metabolic pathways of grain-related traits through protein degradation, synthesis, and targeting secretory pathways. 

### 2.7. Expression Profile of CGs

The RGAP database demonstrates the FPKM expression values obtained for different tissue, including leaves at 20 days, post-emergence inflorescence, pre-emergence inflorescence, anther, pistil, seeds 5 days after pollination (DAP), embryos (25 DAP), endosperm (25 DAP), seeds (10 DAP), and shoots. Expression values of all the 60 CGs in specific tissues were illustrated as heatmap (Figure 9, Appendix A). The size and quality of rice grains are shaped by the conjoined growth of maternal and zygotic tissues. Cell expansion acts as an intrinsic force, propelling longitudinal elongation of the caryopsis within the first 6 days after pollination (DAP), while lateral expansion predominantly takes place between 3 and 8 DAP (Ren et al., 2023). *LOC_Os05g06480.1*, *LOC_Os05g06430.1*, *LOC_Os04g58280.2*, *LOC_Os04g58240.1*, *LOC_Os06g06030.1*, *LOC_Os06g06090.1*, *LOC_Os04g39560.1*, *LOC_Os03g53110.1*, *LOC_Os03g53100.1*, *LOC_Os05g06460.1*, *LOC_Os05g06440.1*, and *LOC_Os02g42700.1* had higher expression in seeds (5 DAP) and seeds (10 DAP), suggesting an association between CGs and the grain size and quality traits. CGs *LOC_Os01g07760.2*, *LOC_Os01g07780.1*, *LOC_Os03g03020.1*, *LOC_Os03g03130.1*, *LOC_Os03g07450.1*, *LOC_Os05g40820.2*, *LOC_Os05g40740.3*, *LOC_Os08g43470.1*, and *LOC_Os08g43540.1*, associated with trait aroma, had higher expression in post-emergence inflorescence and pre-emergence inflorescence, the pistil, and in seeds (5 DAP). Post-emergence inflorescence refers to the stage in a plant’s development when the flowering structures (such as flowers, spikelets, or florets) become visible after the vegetative growth phase. Aroma volatiles are synthesized in the aerial parts of the plant after the development of flowering structures and deposited in mature grains [43]. Therefore, high expression of CGs in post-emergence inflorescence and pre-emergence inflorescence might indicate a potential association between these candidate genes and aroma volatiles being formed.

### 2.8. Exploration of Four Potential QTNs Associated with Grain-Related Traits

We selected four QTNs (*qGL-5-1*, *qGW-6-1*, *qAro-8-4*, and *qLWR-3-2*) anticipated to significantly impact grain size and quality traits, guided by gene ontology and the expression levels of candidate genes (CGs), and proceeded to investigate them further. The candidate region of 3.28 Mb to 3.33 Mb in *qGL-5-1* was defined in the LD block (Appendix A), considering a threshold value of r^2^ > 0.2. Three genes in this genomic region were potential CGs governing grain length in rice. qGL-5-1 was found to be associated with *LOC_Os05g06470*, which is a suppressor of Mek (MAP kinase kinases or mitogen-activated protein kinase kinases) [62]. The products of mitogen-activated protein kinase (MAPK) cascades are determined via phosphorylation of MAPK substrates. There are several studies in which the MAPK signaling pathway for the control of grain size have been reported [31,63]. *LOC_Os05g06450* (Tubulin/Ftsz domain-containing-protein—*OsTUG)* neighboring *LOC_Os05g06470*, plays an important role in determining the location of cell division, promoting nuclear separation, and is expected to be essential for microtubule organization [17]. Therefore, *LOC_Os05g06470*, associated with the suppressor of Mek, can play role in grain length attributes in rice.

The QTN for grain width, *qGW-6-1*, located at 2804963 bp at chromosome 6, shows peak association with grain width via the pLARmEB and ISIS EM-BLASSO methods, with an LOD score of 3.82. A total of 87.15 Kb (2.74 Mb–2.83 Mb) block was generated (Appendix A). Five candidate genes were found to be associated with grain width. The CGs for this block were *LOC_Os06g06030* (peptidase, T1 family), *LOC_Os06g06050* (OsFBL27—F-box domain and LRR containing protein), *LOC_Os06g06090* (CGMC_MAPKCMGC_2_ERK.12—CGMC includes CDA, MAPK, GSK3, and CLKC kinases), *LOC_Os06g06100* (dihydroneopterin aldolase), and LOC_Os06g06150 (zinc finger, C3HC4-type domain-containing protein). Gene annotation studies have shown their role in grain size development [58,64]. This QTN was also found in the vicinity of *GW6 (LOC_Os06g44100*), present approximately 23.7 Mb downstream. Therefore, *LOC_Os06g06080*, which is associated with serine esterase family protein, can have role in rice grain size development.

The LD block of the 103.89 Kb (27.3 Mb–37.4 Mb) genomic region was mapped with three CGs for a QTN found associated with trait aroma (Appendix A). *qAro-8-4* was found associated with *LOC_Os08g43470*, which is an ER lumen protein-retaining receptor which is responsible for retaining ER lumen proteins from the Golgi apparatus. In a study conducted previously [65], rice mutants were generated by editing three cytochrome P450 (*LOC_Os08g43440* neighboring to qAro-8-4 in the current study) homologs, which exhibited increased grain size and 2-AP in concentration. 2-AP is responsible for grain aroma. It was also demonstrated that gene ontology of the cellular component of rice mutant lines showed enriched membrane components [65]. Since ER is a membranous cellular component, it can have direct or indirect roles in the synthesis of 2-AP. Another locus *LOC_Os08g43430* neighboring our QTN *qAro-8-4* is annotated as CXE carboxylesterase, which belongs to the alpha/beta-hydrolases superfamily containing 331 amino acids. CXE carboxylesterases are reported to positively regulate the catabolism of volatile esters in pear fruit and peach, enhancing the aroma and taste of the fruit [66]. Therefore, *qAro-8-4* was found to be associated with *LOC_Os08g43470* and can have role in the pathways 2-AP synthesis, which is a volatile organic compound responsible for aroma. 

QTN for LWR (*qLWR-3-2*) was found to be associated with *LOC_Os03g53110*, which encodes a Cor-A-like magnesium transporter. Magnesium plays role in rice grain yield and 1000 grain weight [67]. In wheat, magnesium transporters are important for starch distribution and increase in grain size [68]. All the neighboring genes near *qLWR-3-2* have been reported to contribute to grain-size-related traits. Six CGs were identified in the genomic region of 127.25 Kb (Figure 10). *LOC_Os03g53050* (probable WRKY transcription factor 21) acts as a downstream receptor of the MAPK module, thereby regulating the grain size by these modules. *LOC_Os03g53070*, annotated as prenylated rab acceptor, regulates the vesicle trafficking of GTPases. *LOC_Os03g53080* zinc finger C3HC4-type domain-containing protein belongs to the RING finger protein family reported to be involved in the determination of grain size in Arabidopsis and rice [64]. For instance, in Arabidopsis, *DA1*, a ubiquitin receptor, negatively regulates organ size and grain size by restricting the period of cell proliferation [69], whereas RING finger protein *BIG BROTHER (BB*) is a repressor of plant organ size development. Small changes in *BB* expression levels substantially alter organ size, whereby mutation in genes like *EOD1* (an enhancer of *da1-1)/BB (Big Brother*) and *DA2*, which encodes *RING-type E3 ligases*, enhances seed size [69,70,71]. A decreased grain size1 (*dgs1*) mutant was also studied by Zhu et al. in rice [64] and showed reduced grain size compared to the wild type. Furthermore, a previously known gene, *OsGW2*, encoding RING protein, also negatively regulates grain size, which is a RING-type ligase [26]. Other co-localized loci include *LOC_Os03g53100*, and *LOC_Os03g53150*. *LOC_Os03g53100* (response regulator receiver domain) highly expressed in seeds (5 DAP), specifically, is a signal transducer involved in the His-to-Asp phosphorylation relay signal transduction system. It activates type A response regulators in response to cytokinin, which affects cell expansion and proliferation by regulating the cell cycle [72]. *LOC_Os03g53150* is associated with *OsIAA13*; it is an auxin-responsive Aux/IAA gene family member containing 237 amino acids involved in the auxin signaling pathway that modulates the grain growth, cell expansion, cell fate via signal transduction, auxin coupling and auxin catabolism. The auxin-deficient mutant *tsg1*(tillering and small-grain 1) resulted in reduced grain size, whereas overexpression of *BG1* (Big Grain 1) resulted in larger grains due to the alteration in hull cell proliferation as compared to wild types [31,73]. This suggests that chromosome 3 is a hotspot of grain size regulatory genes, forming a cascade which modulates the seed length–width ratio. Hence, the LD block for QTN *qLWR3-2* (*LOC_Os03g53110*) has been shown, which forms the major QTN for LWR.

## 3. Discussion

Finding genomic regions linked to quantitative traits before utilizing them in practical breeding to improve specific traits is essential. Enhancing grain size in rice has captured researchers’ interest due to its significant influence on yield. Several studies have aimed to map the genomic regions controlling these traits and identify the genes involved [6,74]. Connecting gene-based markers to genomic regions controlling grain size traits holds substantial promise as it can address multiple traits simultaneously [75]. Genome-wide association analysis using diverse germplasm accessions offers numerous advantages over traditional bi-parental mapping for identifying QTLs [76]. In our study, the use of rice northeast core and rice landraces collected from the different states of India has been an added advantage for diversifying the GWAS panel. The rich genetic diversity present in rice landraces and northeast core collection establishes it as a significant repository of genetic variability and a potential reservoir of beneficial alleles for rice breeding. Due to limited studies on grain-size- and grain-quality-related traits, there is abundant scope to explore the natural variation that exists in germplasm accessions for grain-quality-related characters. Aroma, on the other hand, comes from volatile organic compounds, one of them being 2-AP, but this is influenced by many factors such as temperature, soil type, etc. [44]. Some studies have revealed a single recessive gene to be responsible for aroma; others have discovered a dominant gene [44,45]. Different methods of screening traits of interest have been used by researchers. As an alternative to the classical markers or DNA/molecular markers method, GWAS has become very prominent and useful. It is exceedingly challenging to explore and harness exotic genes across large collections; hence, the northeast core and rice landrace collection serves to ease this. In this study, 483 genotypes with high potential constituted the association panel and were evaluated for grain size parameters and aroma. After high-throughput genotyping, significant associations were analyzed. 

### 3.1. Phenotypic Variation and Trait Correlation

Analysis of variation, skewness, and kurtosis outcomes supports the constitution of the association panel aimed at detecting potential QTNs via marker–trait associations for grain-related traits utilizing GWAS (Figure 1, Table 1). In the present study, there was considerable phenotypic diversity observed with respect to grain size traits within the population, implying a plentiful presence of allelic variations. Positive and strong correlation was observed between GL and GW, which was consistent with previous studies (Ponce et al., 2020 [9]). Negligible skewness or zero skewness was observed, with aroma and grain size parameters similar to the results obtained previously [6].

### 3.2. Population Structure, Genetic Relatedness and LD Decay Analysis

The population structure analysis revealed the presence of three subpopulations within the GWAS panel of 483 rice genotypes. This finding was consistent with the observation from PCA and NJ tree, which also identified three distinct groups. These findings align with the results reported in previous GWAS studies [77,78,79]. These sub-populations are thought to arise from allelic sharing, attributed to the accumulation of alleles over time due to spontaneous mutations [80]. Finally, the kinship matrix, displayed as a heat map with relatedness values ranging from −2.0 to +2.0, indicates weak relationships among individuals in the association panel. These findings aided in understanding the population structure before embarking on the GWAS to identify potential genomic regions associated with grain-related traits. It is essential in the GWAS that genetic structure analysis must be conducted prior to association, regardless of the type of population, to give an overview of the association panel. LD decayed rapidly with the increase in physical distance between SNP pairs and reached its half maximum around ~130 kb for the GWAS panel. Studies have shown that LD decay distances from 100 kb and over [81] are best suited for association studies. The effectiveness and precision of association mapping rely on the level of LD within the population being studied. A self-pollinating species such as rice, where LD spans around 100 kb and beyond, is exceptionally well suited for GWAS [82]. In the current study, LD was calculated as 133.46kb, which goes beyond 100 kb, suggesting a well-suited GWAS panel.

### 3.3. Dissecting Candidate Genes around the Stable QTNs Identified

Utilizing GWAS for mapping marker–trait associations also enables the discovery of beneficial combinations of alleles concerning trait expression. This aspect is crucial for breeding programs focused on incorporating advantageous alleles in crop enhancement efforts [83]. Association analysis with five multi-locus methods was performed with 483 rice germplasms, and 40 QTNs were identified for four grain-related traits (GL, GW, aroma, and LWR). 

The GWAS using single-locus models, despite the frequent use of such models, comes with drawbacks like a notably high false-positive discovery rate. Correcting for multiple testing, often using the Bonferroni correction factor, becomes necessary, yet this approach has been criticized for being stringent, leading to the dismissal of associations that could actually be valid [84]. Conversely, multi-locus GWAS models offer solutions by mitigating the need for multiple corrections and providing more accurate estimations of QTN effects [85]. Enhancing grain size in rice has captured researchers’ interest due to its significant influence on yield. Several studies have aimed to map the genomic regions controlling these traits and identify the genes involved [6,75]. In the current study, out of the 40 QTNs identified, 16 QTNs i.e., *qGL-4-1*, *qGL-5-1*, *qGL-12-1*, *qGW-2-1*, *qGW-4-1*, *qGW-6-1*, *qGW-12-1*, *qAro-1-2*, *qAro-3-1*, *qAro-3-2*, *qAro-5-1*, *qAro-8-1*, *qAro-8-3*, *Aro-8-4*, *qAro-10-1*, and *qLWR-3-2* were detected with more than one of the ML-GWAS methods. Some of them are discussed as follows. *qGL-5-1* was found associated with *LOC_Os05g06470*, which is a suppressor of mek (mitogen activated protein kinase kinase), putative and expressed. The products of MAPK (mitogen activated protein kinases) cascades are determined via the phosphorylation of MAPK substrates. Earlier, in a study, it was shown that *SMG1* gene coding for *MKK4* influences rice grain size by promoting cell proliferation [63]. Loss of *OsMKKK10* function results in small, light grains and semi-dwarf plants with short panicles [28]. This QTN has also shown a positive allelic effect on GL (Table 2). Among the 16 QTNs, *qGW-2-1* was confined to *LOC_Os02g42600*, which is an RNA-binding motif. *La* proteins in rice were characterized previously as RNA-binding proteins [86]. The mutant form of these *La* proteins showed reduced grain length and pollen fertility. *qGW-4-1* demonstrated a significant association confined to *LOC_Os04g58320*, which is a gene in rice which encodes for a zinc-finger-RING-type, putatively expressed protein. Zinc finger RING proteins are a type of RING finger protein that have a conserved RING domain and mainly function as E3 ubiquitin ligases. In rice, this domain is found to play role in various processes, like regulating grain size [64] and salt tolerance [87]. *qGW-12-1*, identified by two ML-GWAS methods, corresponds to *LOC_Os12g25200*, which is a chloride transporter. The chloride channel family negatively regulate salt tolerance in rice [88]. However, a member of the chloride efflux transporter is involved in mediating grain size as well [89]. A neighboring gene, located 16 kb downstream of *LOC_Os12g25200* (*qGW-12-1*), is *LOC_Os12g25210*, which is a signal-peptidase complex subunit 1. Signal-peptide peptidase is a multi-transmembrane aspartic proteinase involved in regulated intramembrane proteolysis, which is implicated in fundamental life processes such as immunological response, cell signaling, tissue differentiation, and embryogenesis. It can play a role in rice grain width, which requires further analysis [90]. Moving further among the 16 QTNs, for aroma, *qAro-8-4* was identified as an association, which belongs to the ER lumen protein-retaining receptor (*LOC_Os08g43470*). The ER lumen protein-retaining receptor manages the retention of endoplasmic reticulum proteins within the ER’s lumen. It plays a crucial role in determining the specificity of this retention system and is essential for the smooth movement of vesicles through the Golgi apparatus. The gene ontology analysis of cellular components in rice mutant lines revealed enrichment in membrane components. Given that the endoplasmic reticulum (ER) is a membranous cellular component, it may play direct or indirect roles in the synthesis of 2-AP. The role of membrane components in regulating aroma volatiles is worthy of comprehensive studies and confirmation. A neighboring gene within the LD of *LOC_Os08g43470* (*qAro-8-4)* is *LOC_Os08g43440*, which is related to cytochrome P450; through mutations involved in a homolog of P450, it has been reported to generate high yield and improved aroma in rice; therefore, this can serve as a candidate gene for rice aroma [65]. Another neighboring gene near *qAro-8-4* is *LOC_Os08g43430*, related to CXE carboxyl esterase. Carboxyl esterase is positively co-related with the catabolism of volatile esters in pear fruit and enhances their aroma quality [66]. Therefore, *LOC_Os05g06470*, *LOC_Os06g06080*, *LOC_Os08g43470*, and *LOC_Os03g53110* have been proposed as the CGs for GL, GW, grain aroma, and LWR. 

### 3.4. Favorable/Superior Allele Analysis

GWAS is an important tool for detecting favorable alleles for various traits in many plants [91]. The association study resulted in the identification of associations and, in turn, superior alleles, which may play a key role in modulating the agronomic traits of rice. In our study, SNP AX-95936094 was present to 30% in the northeast core set but showed a positive effect for the GL trait. However, the potential molecular functions of the significant markers need to be researched. Likewise, SNPs AX-95946823 for GW, AX-95918031 for aroma, and SNP AX-95918592 for LWR showed low percentages of superior alleles. Sometimes, individual markers may account for only minor phenotypic variance, but the aggregation of favorable alleles from diverse marker loci into a single recipient parent can yield significantly greater effects, potentially leading to the development of elite cultivars [92]. We anticipate that these superior alleles through proper enhancement could significantly impact grain-related traits and provide valuable insights for breeding and enhanced rice varieties in the future.

### 3.5. Enrichment Analysis of Identified Candidate Genes

In the current study, GWAS revealed 258 genes for four traits based on the gene expression data and annotations. Of these, 60 genes showed higher expression in specific tissues and were predicted to be involved in pathways affecting grain quality. Gene functional enrichment analysis showed that the GO term “regulation of biological quality” (GO:0065008, FDR = 1.30 × 10^−11^) was significantly enriched, indicating that the overlapping genes modulates a qualitative or quantitative trait of a biological quality (Appendix A). The overlapping genes include *LOC_Os08g43470* (*qAro-8-4*), which was found to be significantly associated with trait aroma. The GO:0051173 term, associated with the positive regulation of nitrogen compound metabolic processes, coincided with *LOC_Os10g10990*, indicating its potential involvement in influencing aroma traits. This correlation is supported by prior research demonstrating that increased nitrogen application enhances aroma in rice [56]. Additionally, molecular and cellular components were annotated and found to be enriched in the 60 candidate genes (CGs). Analyzing trait ontology is a productive approach for exploring the connections between genes and traits. *LOC_Os05g06480* (encoding inorganic H+ pyrophosphatase), *LOC_Os06g06050* (coding for F-box/LRR-repeat protein), and *LOC_Os06g06090* (related to mitogen-activated protein kinase 6) were identified as being linked to the enriched TO terms. All three are shown to be related to grain size directly or in associated ways. Hence, trait ontology (TO) serves as a valuable tool for the systematic exploration of molecular mechanisms that underlie agronomic traits. Genome-wide maps of chromatin states have become a powerful representation of genome annotation and regulatory activity. Chromatin functions as a platform for organizing the genome, overseeing gene expression, cell division, differentiation, and more. Epigenetic regulation, including DNA methylation, histone modifications, and variants, plays a pivotal role in governing chromatin structure. The interplay of various epigenetic mechanisms can give insight into the regulatory roles of genes. In our study, CGs were found to be associated with epigenetic markers (H3K36me1, H3K36me2, H3K36me3, and H3K4me3) of gene activation. Epigenetic elements, encompassing chromatin histone modifications, DNA alterations, and miRNA regulation, operate independently of changes in DNA sequence. Reports have highlighted the role of epigenetic mechanisms in regulating grain size in rice and other plant species. For instance, *RAV6* encodes a *B3* DNA-binding domain protein, and heightened expression of this gene correlates with smaller grains, regulated, in part, by methylation [93]. The semidominant mutation *Epi-rav6*, which mimics *RAV6* overexpression, enhances leaf inclination and grain size by impacting BR (Brassinosteroid) homeostasis. BR signaling via ubiquitin pathway has shown to play direct role in grain size [58]. Thus, chromatin states, together with gene ontology and trait ontology, indicate the CGs having role with different degrees of activity in grain size development and grain aroma. Finally, the presence of a nearby LD block encompassing the robust QTNs *qGL-5-1*, *qGW-6-1*, *qAro-8-4*, and *qLWR-3-2*, along with the surrounding candidate genes, indicates a relatively stable heritability for their associated traits, potentially unaffected by LD block effects.

## 4. Materials and Methods

### 4.1. Association Mapping Panel

A diverse set of 483 rice germplasms comprising 190 accessions from the northeast core set of India, comprising the states of Tripura, Manipur, Nagaland, Assam, Meghalaya, Mizoram, and Arunachal Pradesh, and 293 rice landraces from other different states of India viz. Uttar Pradesh, Jharkhand, Chhattisgarh, Andaman, West Bengal, and Uttarakhand were utilized in the study. These diverse sets of germplasms constituted the association panel for the GWAS for the identification of significant marker–trait associations for GL, GW, grain aroma, and LWR. The details of genotypes are presented in Appendix A.

### 4.2. Phenotyping and Phenotypic Analysis

Seeds were procured from the gene bank in two sets, with fifty grains in each set, for evaluation of grain quality traits. One set of seeds underwent dehusking and milling in the laboratory using a rice husker and milling machine (model JGMJ 8098, Made-in-China, Nanjing, China) following the cleaning of the paddy to achieve the optimal moisture level. Three grain-related traits, GL, GW, and LWR, were measured using a digital scanner, Biovis PSM Seed Analyzer (Limburg an der Lahn, Germany). The next set of fifty grains, after dehusking and milling, was used for the evaluation of aroma. The aroma was evaluated using the sensory KOH method [94]. In the analysis, two fragrant Basmati rice varieties, namely, Pusa-1121, given an aroma score of 3, and (Pusa basmati-1) PB-1, given an aroma score of 2, along with a non-aromatic rice variety, Pusa-44, with an aroma score of 0, were employed [46]. Each sample underwent evaluation by seven experts to verify the phenotype. Simultaneously, the range, mean value, deviation, and phenotypic coefficient of variation for each trait were computed using R. The interrelationships among quality traits were examined by assessing the linear correlation through the R package psych [95].

### 4.3. DNA Isolation and SNP Genotyping

Eight to ten seeds were carefully placed on 30 × 45 cm seed germination paper, with 2 to 3 cm gaps between them. The paper was properly folded and placed in a germination tray with a water level of up to three centimeters. These trays were then kept in a growth chamber at 28 °C and 90% relative humidity. Rice accessions were grown in batches over two weeks, with accessions from each region processed separately. Young leaves were collected after 15 days, and genomic DNA was isolated from the 483 germplasm accessions using the CTAB method [96]. The DNA quality was assessed on a 0.8% agarose gel and quantified using a Nanodrop spectrophotometer (NanoDropTM 2000/2000c, Thermo Fisher Scientific, Greenville, NC, USA). The 50 K SNP chip, based on single-copy genes and covering all 12 rice chromosomes, was used for genotyping the 483 rice germplasm accessions. The chip provides extensive genome-wide coverage, with an average distance of 0.745 kbp between adjacent SNPs. SNP identification and array design of 50 kSNP chip: Gene-based SNPs were identified from public databases (OryzaSNP, Gramene v6, Raleigh, NC, USA) and through in-house sequence alignment with Bioedit v7.2 and ClustalW 2.1, focusing on 35 bases around each SNP. SNP assays were designed and validated in silico using the AxiomGTv1 algorithm of APT. SNPs with *p*-convert values >0.30 were selected, resulting in a rice Affymetrix chip containing 50,051 high-quality SNPs.

Target probe preparation and 50 K rice SNP array hybridization: Rice genomic DNA was extracted using the CTAB method, quantified with a nano-drop spectrophotometer, and checked on a 1% agarose gel. For target probe preparation, 20 μL of gDNA at 10 ng/μL was used per sample, following the Affymetrix Axiom^®^ 2.0 Assay Manual (Affymetrix Axiom^®^, Thermo Fisher Scientific, Singapore). The process included DNA amplification, fragmentation, chip hybridization, single-base extension, and signal amplification, followed by staining and scanning 

SNP allele calling and data analysis: SNP genotypes were called using the Affymetrix Genotyping Console™ v4.1 software. SNPs with low call rates were excluded, retaining those call rates >95.0% [97].

### 4.4. Phylogenetic Study, Population Structure, Kinship, and Linkage Disequilibrium (LD) Analysis

For final analysis, the stringent filtering strategy was conducted to choose high quality SNPs for association. Markers were imputed using Beagle v4 [98]. Markers with minor allele frequency (MAF) < 0.05 were removed and over 10% missing reads were excluded from the analysis. The final count of markers stood at 35,286. The neighbor-joining tree was constructed based on the SNP data using TASSEL v5.2.82 [99] software and visualized using the interactive tree of life (iTOL) software v6 [100]. The number of subgroups in the association mapping panel was estimated using both a model-based approach using STRUCTURE 2.3.4 software [101] and principal component analysis. To infer the value of genetic cluster (K), each individual was run from K = 1 to K = 10 with 3 iterations for each population. For each run, 100,000 burn-in steps followed by 100,000 Markov chain Monte Carlo simulations were implemented. The optimum number of K was determined according to Evanno et al. (2005) [102], embedded in the structure harvester. PCA analysis, which was incorporated in the package “Adegenet v2.1.10’ [103] (genomic association and prediction integrated tool) running under R environment, was used. PCA was also used to infer population structure. The first two PCs were plotted using ggplot2 in R 3.4.2 to visualize the dispersion of the rice accessions. The LD analysis was performed via pairwise comparisons in a set of 35,286 SNP markers (MAF < 0.05) using the LD function in TASSEL v.5.2.82 [99]. The value on the x-axis (distance bp), where r^max^ (y-axis) was dropped to half, was calculated to be the LD. 

### 4.5. Association Analysis and Identification of Potential Candidate Genes

A GWAS analysis was conducted on 483 rice accessions utilizing 35,286 high-quality SNPs with default settings of in mrMLM software v4.0.2 to estimate the significant associations for grain length, grain width, grain length–width ratio, and grain aroma. Five multi-locus models, namely, mrMLM, FASTmrMLM, FASTmrEMMA, pLARmEB, and ISIS EM-BLASSO, were utilized using R package to pinpoint candidate QTNs (https://cran.r-project.org/web/packages/mrMLM/index.html (accessed on 22 November 2023). To mitigate potential false positives due to population structure, the analysis incorporated the first three principal components (PCs) and a kinship matrix as covariates within the framework. Considering an LOD score value ≥3 as the threshold, significant QTNs were identified [104]. The common QTNs detected by any two ML-GWAS models were predicted to be good candidates for rice grain related traits. All genes situated within the LD decay distance of the identified QTNs were extracted and underwent comprehensive gene annotation studies to find candidate loci for each trait, employing the Rice Annotation Project-Database (RGAP, http://rice.uga.edu/) [105].

### 4.6. Candidate Gene Analysis and Gene Functional Enrichment Analysis

The Rice Genome Annotation Project Database (RGAP, MSUv7.0, http://rice.uga.edu/) [105] and Information Commons for Rice (IC4R, http://ic4r.org/) were used to search and functionally annotate the putative genes underlying the ±130 Kb genomic region of the common significant QTNs identified via various ML-GWAS models. Candidate genes were selected based on following criteria: (a) neighboring genes of significant QTNs lying within the LD decay distance; and (b) genes with a known function in rice or Arabidopsis orthologs associated with traits of interest. Subsequently, gene ontology (GO) enrichment analysis was conducted on 60 putative candidate genes, with the aim of acquiring better insight into the molecular and biological role, using the Plant Gene Set Annotation Database (PlantGSAD) analysis tool [106] and analyzing the data with REVIGO [55]. For further analysis, trait ontology (TO) categories, pathway gene sets (Mapmam gene set), and chromatin states were selected for gene set enrichment analysis using PlantGSAD. TO terms, pathways, and chromatin states with a *p*-value less than 0.05 were regarded as significantly enriched, indicating that the set is enriched with the genes of a particular pathway or functional category.

### 4.7. In Silico Gene Expression Analysis of Candidate Genes

The expression values available at the Rice Genome Annotation Project Database (RGAP, MSUv7.0, http://rice.uga.edu/) and the Rice Expression Database (RED, http://expression.ic4r.org/) were utilized to investigate all the candidate genes in different tissues to further depict the associations between genes and phenotypic traits. Annotated genes were compared with their homologs using the Arabidopsis Information Resource (TAIR) database (https://www.arabidopsis.org/). The R package “heatmap” was used to create a heatmap depicting the FPKM values of the candidate gene. Genes with the elevated expression in specific tissues and putative or known functions associated with desired traits were identified and further investigated. A schematic representation of the methodology followed in our study is shown in Figure 11.

### 4.8. LD Block Analysis

Four candidate genes associated with traits of interests were further investigated. The LD block of four selected robust loci were generated using filtered SNPs following the established confidence interval by Gabriel [107]. LD heatmaps were created using the LD Block Show tool v1.39. All genes situated within the LD decay range of the identified QTNs were mined and subjected to comprehensive investigation.

## 5. Conclusions

In the current study, five ML-GWAS models were employed for grain-related traits on a set of 483 rice GWAS panels, with 190 accessions from northeast core and 293 from rice landraces, using 35,286 SNPs. The number of QTNs identified were 8, 9, 12, and 11 for GL, GW, aroma, and LWR, respectively. Amongst the 40 different QTNs in total, 16 were obtained with two ML-GWAS methods simultaneously. We examined all 16 genomic loci linked to grain quality traits in Rice Assembly version 7 and annotated them accordingly. Probable candidate genes (CGs) were sought within the 130 kbp genomic region surrounding each of the 16 commonly annotated QTNs. Across these 16 QTNs, 258 genes were identified as being in close proximity to significant QTNs. Among them, 60 genes exhibited elevated expression levels in specific tissues, as indicated by the available FPKM values from the RGAP and RED databases, and were predicted to play roles in pathways influencing grain quality. Meanwhile, we also studied the superior allele in the northeast core set and the rice landrace set. Some superior SNPs have been shown to be present to less than 30% of the genotypes, indicating the need to uncover their potential molecular function, which would be beneficial to pyramid breeding. Subsequently, gene annotation, gene ontology, trait ontology, and enrichment analysis showed that 60 CGs were found to be enriched, in GO terms, in the studied traits, and they also showed higher expression in seeds (5 DAP) and seeds (10 DAP), suggesting an association between CGs and the grain size and quality traits. *LOC_Os05g06470*, *LOC_Os06g06080*, *LOC_Os08g43470*, and *LOC_Os03g53110* were confirmed as key candidates by expression analysis, GO, and T, as well as pathways analysis for aforementioned traits. Choosing elite genotypes identified for their higher occurrence of desirable alleles linked to grain size traits and aroma could accelerate the pace of rice enhancement, tackling issues concerning food security and sustainable rice cultivation. Moreover, from a breeding point of view, forty MTAs exhibiting significant associations with grain-related traits hold promise for gene cloning, which, in turn, can be leveraged for marker-assisted selection (MAS) aimed at enhancing grain-related traits. The MTAs uncovered in the current study are pivotal as they may be linked to minor genes influencing target traits. Utilizing SNP markers associated with specific loci, favorable alleles can be stacked in newly emerging varieties to enhance their traits. Furthermore, these varieties have the potential to serve as valuable parents in breeding programs aimed at enhancing grain quality parameters through genetic enhancement. GWAS models have uncovered genetic variants linked to various traits, a phenomenon known as pleiotropy. This identification of pleiotropic loci holds significance in comprehending the common origins of diseases and complex traits. Thus, candidate genes can help in MAS and precision breeding, whereby traits like GL can be precisely targeted without affecting other traits. CGs can help in pyramiding the genes, whereby breeders can leverage this knowledge to pyramid multiple favorable alleles into elite rice varieties, leading to further improvements in grain length and other related traits, such as grain weight and yield. Furthermore, the potential candidate genes identified are crucial targets for future studies aimed at functional characterization. Such research endeavors can help bridge the gaps within and/or construct a genetic framework for signaling pathways that regulate grain size and aroma in rice.

## Figures and Tables

**Figure 1 plants-13-01707-f001:**
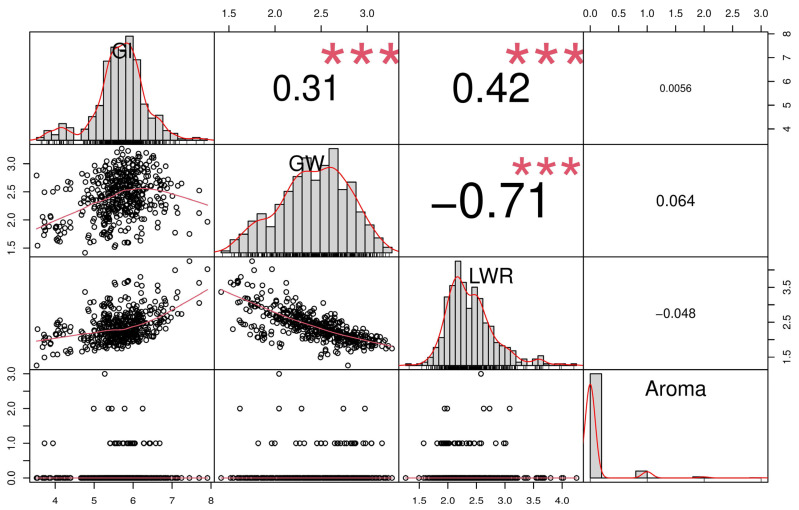
Correlation coefficient matrix, scatter plot, and phenotypic frequency distribution among grain-related traits. Each variable’s distribution is displayed diagonally. The bivariate scatter plots with a trend line are shown at the bottom of the diagonal. The correlation coefficient and the level of significance are displayed as stars at the top of the diagonal (*** *p* > 0.001 shows significance level).

**Figure 2 plants-13-01707-f002:**
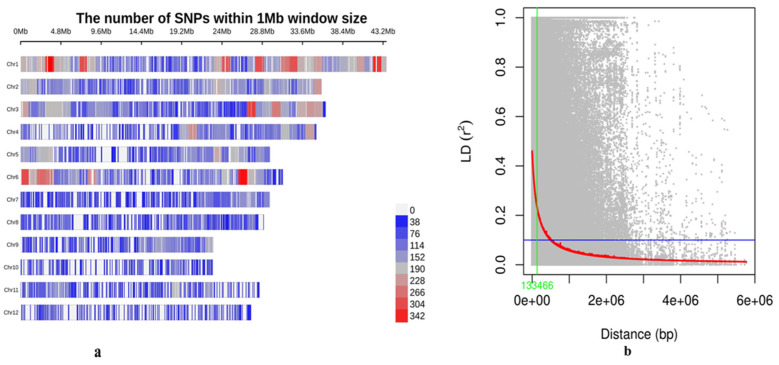
SNP marker distribution and LD decay of 483 rice accessions: (**a**) The distribution of SNPs within 1 Mb window size across 12 rice chromosomes; (**b**) LD decay distance in the whole population. Pairwise LD (r^2^) values against the corresponding pairwise physical distance (bp) of SNP markers were plotted. Red line indicates the trend line of non-linear regressions against physical distance.

**Figure 3 plants-13-01707-f003:**
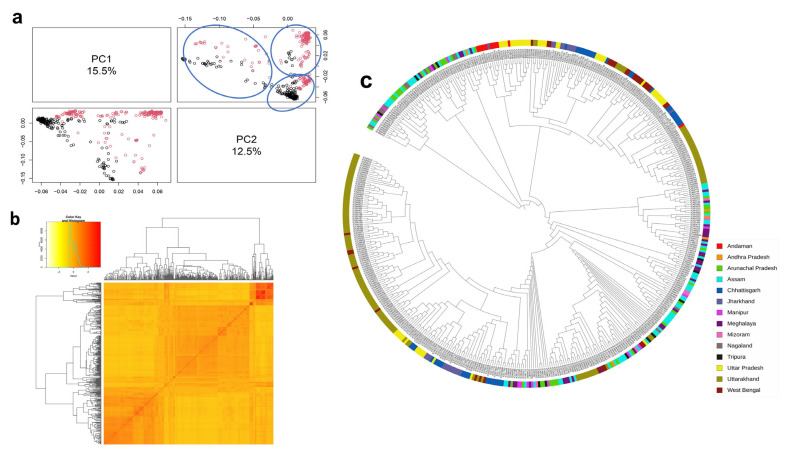
Population stratification and diversity analysis of 483 accessions using 35,286 high-quality SNPs: (**a**) principal component analysis 3D plot (black—190 ne core; red—293 rice landraces; Blue ellipses shows three subclusters among the 483 rice panel) (**b**) heatmap of pairwise kinship matrix; (**c**) phylogenetic tree based on neighbor-joining method.

**Figure 4 plants-13-01707-f004:**
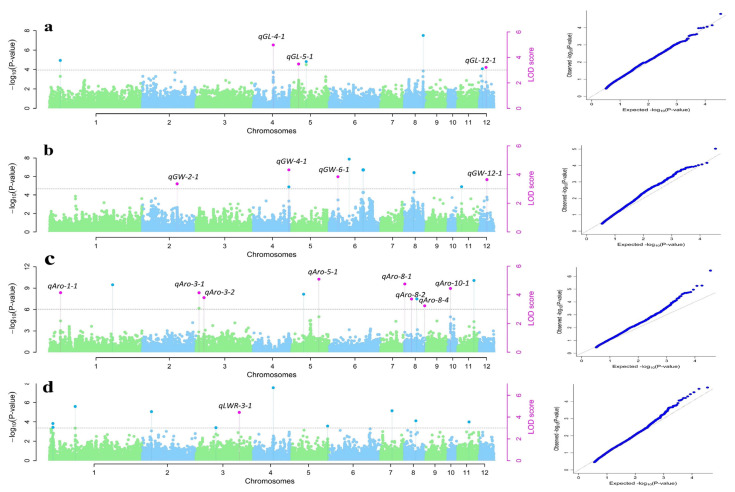
Manhattan plots and quantile–quantile plots for (**a**) grain length, (**b**) grain width, (**c**) aroma, and (**d**) length–width ratio using five multi-locus models. The horizontal dotted line indicates the threshold LOD score ≥3. The dots above the threshold value represent the significant QTNs at different rice chromosomes; the dots in pink color represent QTNs detected by ≥2 models.

**Figure 5 plants-13-01707-f005:**
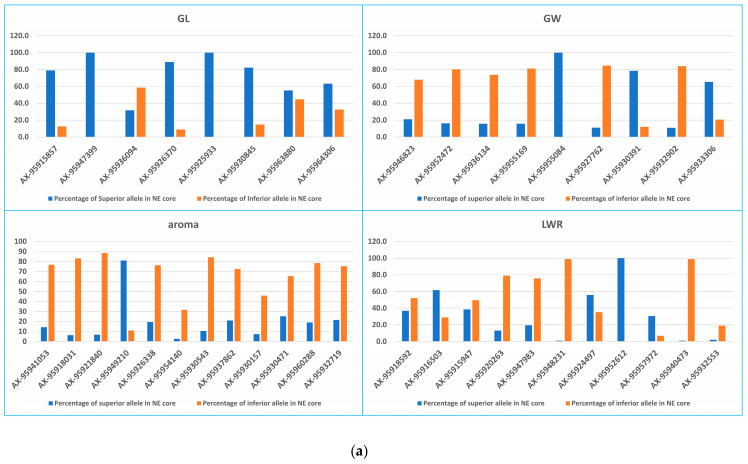
(**a**) Superior and inferior allele distribution in the northeast core; (**b**) superior and inferior allele distribution in rice landraces.

**Figure 6 plants-13-01707-f006:**
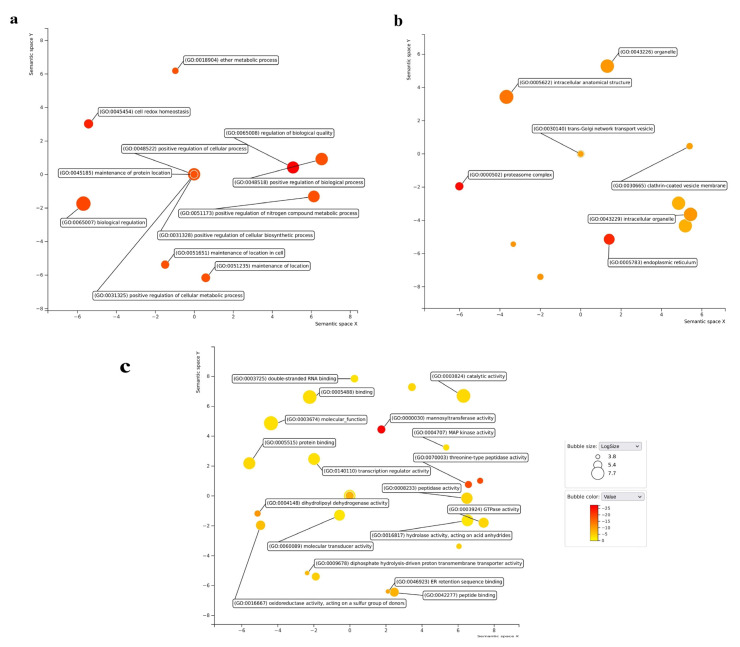
GO enrichment analysis of CGs by PlantGSAD and REVIGO. Scatter plot illustrates the cluster representatives. (**a**) Biological processes; (**b**) cellular component; (**c**) molecular function positioned in a two-dimensional space comprising significant GO terms with semantic similarities. Bubble color and size signify the −log10(*p*) value.

**Figure 7 plants-13-01707-f007:**
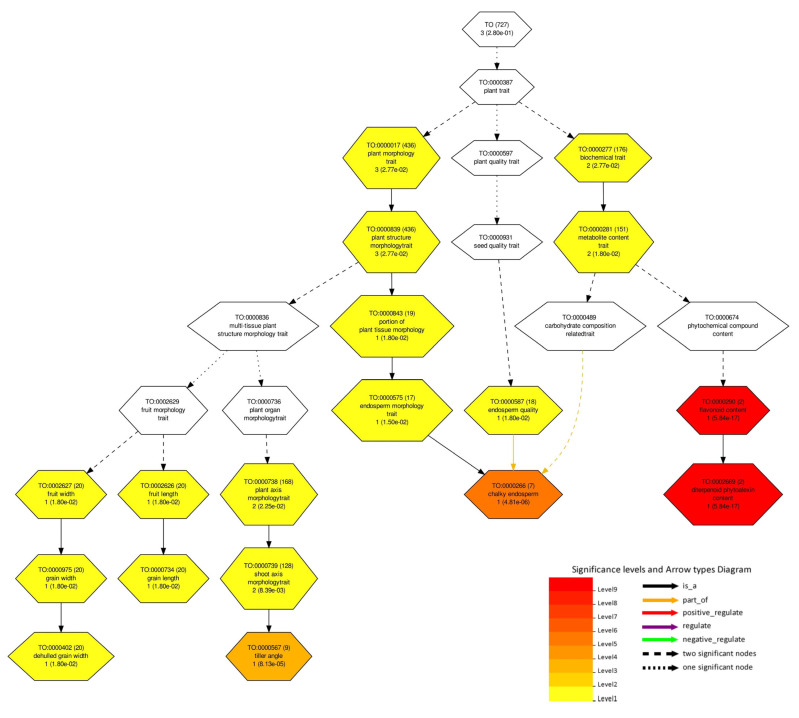
Trait ontology tree depicting the TO terms associated with CGs. Boxes in the diagram represent the TO terms corresponding to a seven-digit ID number preceded by TO, their description, and *p*-value. Colored nodes indicate the significantly enriched TO terms, and arrows indicate the relationship between consecutive nodes.

**Figure 8 plants-13-01707-f008:**
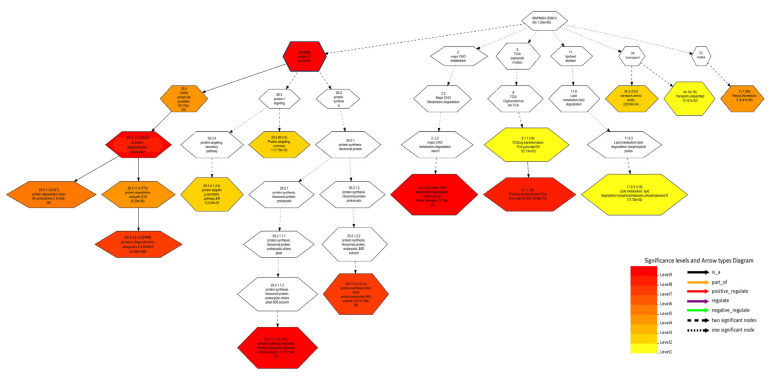
Tree depicting the pathways and processes of MapMan associated with CGs. Boxes in the diagram represent the bincode followed by the description of pathways and *p*-value.

**Figure 9 plants-13-01707-f009:**
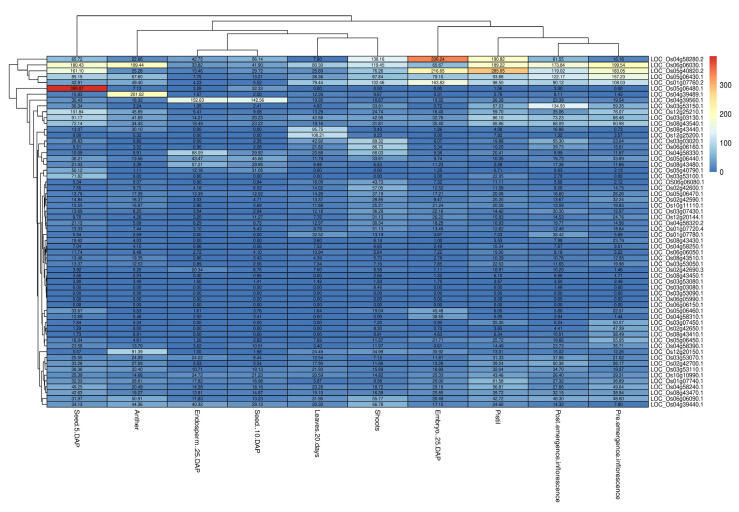
Heatmap showing the normalized FPKM expression values of the 60 CGs for the four grain quality traits.

**Figure 10 plants-13-01707-f010:**
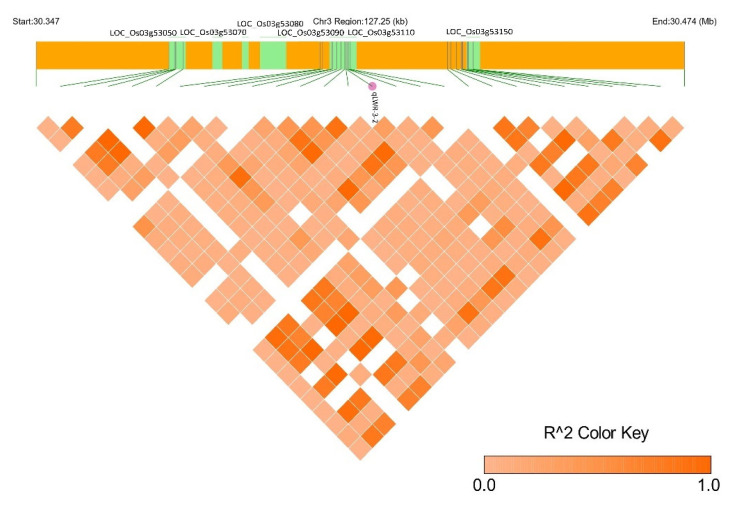
LD block of significant QTN *qLWR3-2* (*LOC_Os03g53110*) associated with grain LWR trait.

**Figure 11 plants-13-01707-f011:**
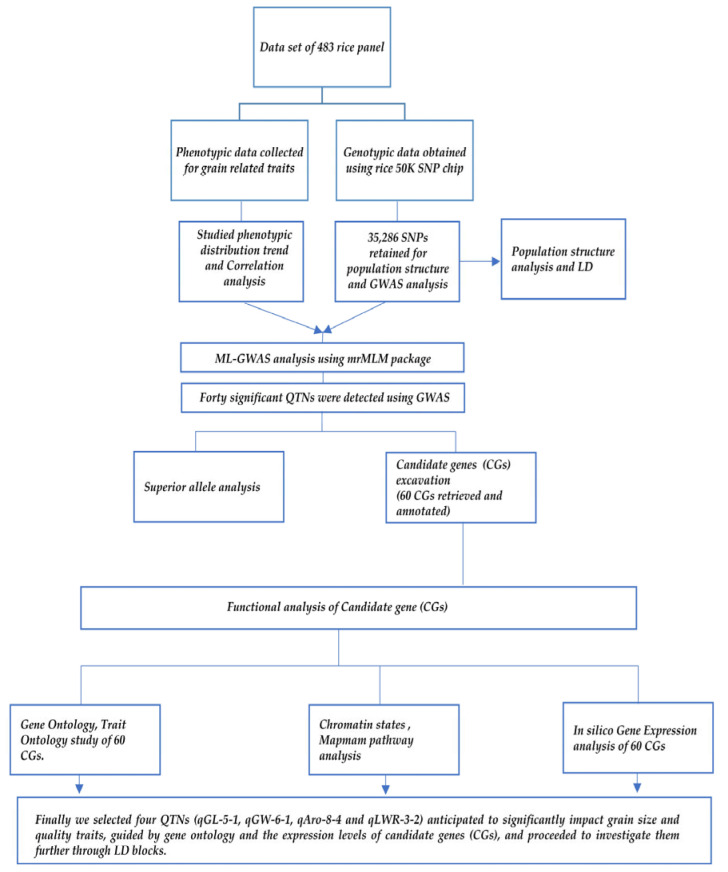
Schematic representation of the methodology followed in our study.

**Table 1 plants-13-01707-t001:** Phenotype variation and distribution pattern of four grain-related traits.

	Mean	SD	Range	Skewness	Kurtosis	CV_Percentage
GL	5.658629	0.682624	4.38594	−0.49934	1.305461	12.0634111
GW	2.428859	0.382422	1.8533	−0.27133	−0.55127	15.74491809
LWR	2.377681	0.427131	2.990061	1.053315	1.789657	17.96418124
Aroma	0.089027	0.338326	3	4.379833	22.1524	380.0265163

**Table 2 plants-13-01707-t002:** Table showing 40 associated QTNs identified for four grain-related traits (16 QTNs identified simultaneously by two or more ML-GWAS methods have been marked in bold).

Trait	Method	Marker name	Chromosome	Marker Position (bp)	QTNs	QTN Effect	LOD Score	r2 (%)	MAF	Genotype for Code 1	Gene IDs	Annotation
Grain Length	mrMLM	AX-95915857	1	3679136	qGL-1-1	0.1913	3.7641	5.6871	0.1975	CC	LOC_Os01g07650	expressed protein
	ISIS EM-BLASSO	AX-95947399	2	3704491	qGL-1-2	0.1371	3.0412	1.8314	0.1284	CC	LOC_Os02g07200	protein-O-fucosyltransferase 1, putative, expressed
	FASTmrMLM/FASTmrEMMA	AX-95936094	4	23306070	qGL-4-1	−0.000026~−0.27	4.5217~5.5504	0.000000145~3.7609	0.4327	GG	LOC_Os04g39510	expressed protein
	FASTmrEMMA/mrMLM	AX-95926370	5	3306008	qGL-5-1	0.2846~2.994	3.4738~3.588	1.4796~1.567	0.1056	AA	LOC_Os05g06470	suppressor of Mek, putative, expressed
	pLARmEB	AX-95925933	5	14277675	qGL-5-2	−0.1238	3.6741	2.5645	0.1366	CC	LOC_Os05g24760	harpin-induced protein 1 domain containing protein, expressed
	mrMLM	AX-95930845	8	26316254	qGL-8-1	0.1895	5.7229	6.0554	0.2422	TT	LOC_Os08g41890	microtubule associated protein, putative, expressed
	mrMLM/pLARmEB/FASTmrMLM/ISIS EM-BLASSO	AX-95963880	12	11729858	qGL-12-1	−0.0734~−0.1123	3.1694~3.2826	1.0612~2.4436	0.3929 ~0.3954	CC	LOC_Os12g20150	phosphoglucan, water dikinase, chloroplast precursor, putative, expressed
	FASTmrEMMA	AX-95964306	12	5033082	qGL-12-2	−0.2342	3.1072	1.4545	0.1946	AA	LOC_Os12g09570	GTP-binding protein, putative, expressed
Grain Width	FASTmrMLM/FASTmrEMMA	AX-95946823	2	25626218	qGW-2-1	−0.0000137~−0.0009	3.355~4.1584	0.0000000636 ~0.0000993	0.1967~0.2536	AA	LOC_Os02g42600	double-stranded RNA-binding motif containing protein, expressed
	FASTmrEMMA, mrMLM	AX-95952472	4	34497326	qGW-4-1	0.1513~0.1655	4.338~4.563	1.5816~1.65	0.1304~0.1415	AA	LOC_Os04g58320	zinc finger, RING-type, putative, expressed
	pLARmEB	AX-95936134	4	34501576	qGW-4-2	0.0664	3.141	1.2942	0.2412	GG	LOC_Os04g58330	expressed protein
	pLARmEB/ISIS EM-BLASSO	AX-95955169	6	2804963	qGW-6-1	0.0000239~0.1504	3.8218~4.9504	0.00000014 ~5.3481	0.1511~0.1511	AA	LOC_Os06g06080	serine esterase family protein, putative, expressed
	pLARmEB	AX-95955084	6	8061158	qGW-6-2	0.0011	5.0742	7.00 × 10^−4^	0.2733	TT	LOC_Os06g14412	-
	ISIS EM-BLASSO	AX-95927762	6	26193259	qGW-6-3	−0.123	4.4299	3.1062	0.1263	GG	LOC_Os06g43560	phox-domain-containing protein, putative
	ISIS EM-BLASSO	AX-95930391	8	25765812	qGW-8-1	−0.0681	5.299	2.4006	0.2019	TT	LOC_Os08g40700	retrotransposon protein, putative, unclassified, expressed
	FASTmrMLM	AX-95932902	11	4372488	qGW-11-1	2.53 × 10^−5^	3.1536	1.18 × 10^−7^	0.0973	GG	LOC_Os11g08300	aldehyde dehydrogenase, putative, expressed
	mrMLM/FASTmrMLM	AX-95933306	12	14486551	qGW-12-1	0.0874~0.0639	3.2978~3.9913	4.2112~2.2787	0.2297~0.2288	CC	LOC_Os12g25200	chloride transporter, chloride channel family, putative, expressed
Aroma	FASTmrMLM/pLARmEB	AX-95941053	1	3733408	qAro-1-2	−0.0826~−0.0664	3.9647~4.3223	5.4883~2.3916	0.323~0.323	GG	LOC_Os01g07780	embryo-specific 3, putative, expressed
	mrMLM	AX-95918031	1	32403084	qAro-1-1	−0.0743	4.695	4.2599	0.3368	TT	LOC_Os01g55700	NLI interacting factor-like phosphatase, putative, expressed
	mrMLM/pLARmEB	AX-95921840	3	1255081	qAro-3-1	0.1321~0.0947	4.642~3.6354	1.6736~0.6132	0.1081~0.1118	TT	LOC_Os03g03080	expressed protein
	mrMLM/FASTmrMLM/pLARmEB	AX-95949210	3	3746297	qAro-3-2	0.0706~0.082	3.6107~5.164	2.2391~ 4.2313	0.2027~0.2019	GG	LOC_Os03g07420	3-dehydroquinate synthase, putative, expressed
	pLARmEB	AX-95926338	5	7909609	qAro-5-1	−0.001	4.0391	6.00 × 10^−4^	0.3489	TT	LOC_Os05g14170	expressed protein
	mrMLM/FASTmrMLM/pLARmEB/ISIS EM-BLASSO	AX-95954140	5	23838482	qAro-5-2	−0.1011~0.1351	4.2957~5.0865	2.9072~4.5818	0.0832~0.084	TT	LOC_Os05g40790	CCR4-NOT transcription factor, putative, expressed
	mrMLM, FASTmrMLM	AX-95930543	8	353775	qAro-8-1	−0.0362~−0.0883	3.2939~ 5.467	1.731~5.266	0.2443~0.2453	GG	LOC_Os08g01600	polygalacturonase, putative, expressed
	mrMLM/FASTmrMLM	AX-95937862	8	7680987	qAro-8-2	−0.0876~−0.0584	3.4328~ 3.9724	6.1051/2.8655	0.317/0.3178	GG	LOC_Os08g12930	hypothetical protein
	mrMLM	AX-95930157	8	19040720	qAro-8-3	0.0676	3.7067	2.5924	0.29	AA	LOC_Os08g31060	phospholipase D alpha 1, putative, expressed
	pLARmEB, mrMLM	AX-95930471	8	27365210	qAro-8-4	2.00 × 10^−4^~3.00 × 10^−4^	3.2194~3.311	1.77 × 10^−5^~1.88 × 10^−5^	0.2246~0.3355	CC	LOC_Os08g43470	ER lumen protein-retaining receptor, putative, expressed
	mrMLM/FASTmrMLM/FASTmrEMMA	AX-95960288	10	5865033	qAro-10-1	0.1023~0.2771	3.0582~5.8835	2.3016~4.7461	0.0759~0.0797	AA	LOC_Os10g11022	expressed protein
	FASTmrEMMA	AX-95932719	11	20143226	qAro-11-1	0.1237	4.9965	2.1561	0.2267	AA	LOC_Os11g35210	NB-ARC domain containing protein, expressed
Length–Width Ratio	FASTmrMLM	AX-95918592	1	1413408	qLWR-1-1	2.76 × 10^−5^	3.408	3.61 × 10^−7^	0.3727	TT	LOC_Os01g03510	WD domain, G-beta repeat domain containing protein, expressed
	FASTmrEMMA	AX-95916503	1	1420092	qLWR-1-3	0.0981	3.0561	1.0236	0.4451	AA	LOC_Os01g03520	ubiquitin conjugating enzyme protein, putative, expressed
	FASTmrMLM	AX-95915947	1	9269345	qLWR-1-2	−1.91 × 10^−5^	4.9725	1.84 × 10^−7^	0.4482	GG	LOC_Os01g16350	hydroxymethylglutaryl-CoA lyase, putative, expressed
	mrMLM	AX-95920263	2	5048675	qLWR-2-1	−0.122	4.5	6.3408	0.2173	CC	LOC_Os02g09790	disease resistance protein RPM1, putative, expressed
	FASTmrMLM	AX-95947983	3	10729510	qLWR-3-1	−3.00 × 10^−4^	3.0226	2.09 × 10^−5^	0.1232	CC	LOC_Os03g19180	GCRP8—Glycine and cysteine rich family protein precursor, putative, expressed
	FASTmrMLM/pLARmEB	AX-95948231	3	30407414	qLWR-3-2	−0.000032475~−0.0003	3.5182~5.3355	0.0000005739~0.000033246	0.4493~0.4499	AA	LOC_Os03g53110	CorA-like magnesium transporter protein, putative, expressed
	FASTmrEMMA	AX-95924497	4	23296090	qLWR-4-1	−0.2448	6.6925	7.0641	0.3882	CC	LOC_Os04g39489	amino acid transporter, putative, expressed
	pLARmEB	AX-95952612	5	28919036	qLWR-5-1	4.47 × 10^−5^	3.1763	6.47 × 10^−7^	0.1656	AA	LOC_Os05g50590	expressed protein
	pLARmEB	AX-95957972	7	21390506	qLWR-7-1	−0.0012	4.5794	6.00 × 10^−4^	0.4472	TT	LOC_Os07g35720	expressed protein
	FASTmrEMMA	AX-95940473	8	17068152	qLWR-8-1	−0.1562	3.6527	2.7457	0.4431	GG	LOC_Os08g28180	PPR-repeat-domain-containing protein, putative, expressed
	FASTmrMLM	AX-95932553	11	15884775	qLWR-11-1	0.004	3.5556	0.0058	0.4917	GG	LOC_Os11g28470	expressed protein

**Table 3 plants-13-01707-t003:** List of 60 candidate genes with their functional annotation.

Trait	Gene	Chromosome	Position (bp)	Functional Annotation	Expressed in	Arabidopsis_Homolog
Grain Length	LOC_Os04g39440.1	Chr4	23493028–23495366	ras-related protein	Seed	AT4G17170
	LOC_Os04g39489.1	Chr4	23523092–23514262	amino acid transporter	Seed	AT5G23810
	LOC_Os04g39560.1	Chr4	23560637–23553703	expressed protein	Seed	AT5G56240
	LOC_Os05g06470.1	Chr5	3323282–3334849	suppressor of Mek	Seed	AT3G06670
	LOC_Os05g06480.1	Chr5	3339817–3335248	inorganic H+ pyrophosphatase	Seed	AT1G15690
	LOC_Os05g06460.1	Chr5	3319717–3313391	dihydrolipoyl dehydrogenase	Seed	AT3G16950
	LOC_Os05g06450.1	Chr5	3312616–3307928	tubulin/FtsZ domain containing protein	Seed	AT3G61650
	LOC_Os05g06440.1	Chr5	3303197–3307713	dnaJ homolog subfamily B member 11 precursor	Seed	AT3G62600
	LOC_Os05g06430.1	Chr5	3298055–3294696	OsPDIL2-1 protein disulfide isomerase PDIL2-1	Seed	AT2G47470
	LOC_Os12g20150.1	Chr12	11744531–11727395	phosphoglucan, water dikinase, chloroplast precursor	Seed	AT5G26570
	LOC_Os12g20144.1	Chr12	11715886–11724620	**	**	**
Grain Width	LOC_Os02g42700.1	Chr2	25683601–25679712	thioredoxin	Seed	AT4G03520
	LOC_Os02g42590.1	Chr2	25619021–25624875	WD-40 repeat family protein	Seed	AT3G15470
	LOC_Os02g42600.1	Chr2	25626748–25636072	double-stranded RNA-binding motif containing protein	Seed	AT4G21670
	LOC_Os02g42690.3	Chr2	25675323–25671123	zinc finger, C3HC4 type domain containing protein	Seed	AT4G03510
	LOC_Os02g42650.1	Chr2	25652321–25659494	expansin precursor	Seed	AT4G28250
	LOC_Os04g58250.1	Chr4	34685203–34680601	protein kinase	Seed	AT3G09010
	LOC_Os04g58280.2	Chr4	34697202–34694304	stem-specific protein TSJT1	Seed	AT4G27450
	LOC_Os04g58320.2	Chr4	34714719–34717281	zinc finger, RING-type	Seed	AT5G01520
	LOC_Os04g58390.1	Chr4	34737984–34742719	allantoinase	Seed	AT4G04955
	LOC_Os04g58240.1	Chr4	34677494–34681940	expressed protein	Seed	AT2G20760
	LOC_Os04g58330.1	Chr4	34721250–34717761	expressed protein	Seed	AT2G20740
	LOC_Os04g58310.1	Chr4	34707954–34706681	CBS domain-containing protein	Seed	AT5G53750
	LOC_Os06g06030.1	Chr6	2774076–2771069	peptidase, T1 family	Seed	AT1G13060
	LOC_Os06g06050.1	Chr6	2780715–2785271	OsFBL27—F-box domain and LRR containing protein	Seed	AT2G42620
	LOC_Os06g06090.1	Chr6	2813004–2806543	CGMC_MAPKCMGC_2_ERK.12—CGMC includes CDA, MAPK, GSK3, and CLKC kinases	Seed	AT2G43790
	LOC_Os06g05990.1	Chr6	2754799–2749783	zinc finger family protein	Seed	AT3G54780
	LOC_Os06g06150.1	Chr6	2849506–2848747	zinc finger, C3HC4 type domain containing protein	Seed	AT5G58580
	LOC_OS06g06080.1	Chr6	2803003–2806286	serine esterase family protein	Seed	AT1G29120
	LOC_Os06g06160.1	Chr6	2855441–2859561	IQ calmodulin-binding motif domain containing protein	Seed	AT2G33990
	LOC_Os12g25200.1	Chr12	14491027–14487396	chloride transporter, chloride channel family	Seed	AT3G27170
	LOC_Os12g25210.1	Chr12	14503903–14503331	signal peptidase complex subunit 1	Seed	AT4G40042
Aroma	LOC_Os01g07740.1	Chr1	3718637–3713687	DEAD-box ATP-dependent RNA helicase 14,	**	**
	LOC_Os01g07760.2	Chr1	3728603–3724314	phospholipase D	Seed, Inflorescence	AT3G15730
	LOC_Os01g07780.1	Chr1	3735829–3736828	embryo-specific 3	Seed, Inflorescence	AT2G41475
	LOC_Os01g07720.4	Chr1	3709016–3705420	dolichyl-P-Man Man-PP-dolichyl mannosyltransferase	Seed, Inflorescence	AT2G47760
	LOC_Os03g03080.1	Chr3	1276308–1276989	**	**	**
	LOC_Os03g03020.1	Chr3	1234476–1232214	L11 domain containing ribosomal protein	Seed, Inflorescence	AT1G32990
	LOC_Os03g03130.1	Chr3	1318244–1321665	ubiquitin-conjugating enzyme	Seed, Inflorescence	AT3G57870
	LOC_Os03g07430.1	Chr3	3771557–3767426	protein kinase domain containing protein	Seed, Inflorescence	AT2G02800
	LOC_Os03g07450.1	Chr3	3786933–3784283	homeobox associated leucine zipper	Seed, Inflorescence	AT1G69780
	LOC_Os05g40820.2	Chr5	23943173–23939897	ribosomal protein L24	Seed, Inflorescence	AT3G53020
	LOC_Os05g40790.1	Chr5	23916372–23923564	CCR4-NOT transcription factor	Seed, Inflorescence	AT1G07705
	LOC_Os08g43410.1	Chr8	27459934–27458307	LRP1	Seed, Inflorescence	AT3G51060
	LOC_Os08g43430.1	Chr8	27475042–27473382	CXE carboxylesterase	Seed, Inflorescence	AT2G45600
	LOC_Os08g43440.1	Chr8	27481168–27479061	cytochrome P450	Seed, Inflorescence	AT4G12320
	LOC_Os08g43450.1	Chr8	27488798–27486168	MYB-like protein 1	**	**
	LOC_Os08g43470.1	Chr8	27497190–27501389	ER lumen protein retaining receptor	Seed, Inflorescence	AT4G38790
	LOC_Os08g43480.1	Chr8	27502289–27504972	zinc finger, C3HC4 type family protein	Seed, Inflorescence	AT3G19910
	LOC_Os08g43540.1	Chr8	27533250–27537106	peptidase, T1 family	Seed, Inflorescence	AT5G66140
	LOC_Os08g43510.1	Chr8	27519462–27522329	thaumatin	Seed, Inflorescence	AT4G38660
	LOC_Os10g10990.1	Chr10	6087844–6079970	transcription initiation factor IIF, alpha subunit domain containing protein	Seed, Inflorescence	AT4G12610
	LOC_Os10g11110.1	Chr10	6144589–6150509	**	**	AT3G28670
Length–Width ratio	LOC_Os03g53050.1	Chr3	30422191–30425543	WRKY121	Seed	AT2G30590
	LOC_Os03g53070.1	Chr3	30436527–30437857	prenylated rab acceptor	Seed	AT1G08770
	LOC_Os03g53080.1	Chr3	30445269–30440115	zinc finger, C3HC4 type domain containing protein	Seed	AT1G06770
	LOC_Os03g53110.1	Chr3	30459072–30453759	CorA-like magnesium transporter protein	Seed	AT5G64560
	LOC_Os03g53100.1	Chr3	30451518–30452854	response regulator receiver domain containing protein	Seed	AT3G04280
	LOC_Os03g53150.1	Chr3	30483289–30480662	OsIAA13—Auxin-responsive Aux/IAA gene family member	Seed	AT3G04730
	LOC_Os03g53090.1	Chr3	30449074–30450181	LRR-kinase protein	Seed	AT4G36180

** No gene ontology annotation found.

## Data Availability

All the data are available within the manuscript and Appendix A.

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
