# Peer review of "Discovering New QTNs and Candidate Genes Associated with Rice-Grain-Related Traits within a Collection of Northeast Core Set and Rice Landraces"

_plants, 2024, doi:10.3390/plants13121707_

Round 1

Reviewer 1 Report

Comments and Suggestions for Authors

In this research, a multi-locus genome-wide association study was conducted on rice genotypes to identify quantitative trait nucleotides associated with grain-related traits. This study discusses insights from QTNs and candidate genes (CGs) that illuminate rice trait regulation and genetic connections, offering potential targets for future studies.

 The analysis of the published data was provided with a sufficient level of scientific novelty. But, there are important flaws in the manuscript listed below:

 -In lines 685 and 686, it is mentioned that "five multi-locus models, namely mrMLM, FASTmrMLM, FASTmrEMMA, pLARmEB, and ISIS EM-BLASSO, were utilized using the R package to pinpoint candidate QTNs". What is the necessity of using five models and what are the advantages of using all these models to pinpoint candidate QTNs?

 -In lines 712 and 712, two criteria are considered for candidate gene selection. It is unclear if both criteria are necessary for selection or if just one of them is enough.

 - In Figure 1, it is unclear what the correlations are between which characters. Further details and explanation are needed.

 - As two sets of accessions (the northeast core set and the set from other states) are being considered in this study (collection of the northeast core set and rice landraces) and various analyses have been conducted, such as in section 2.5, where the distribution of superior alleles in the northeast core and rice landraces was compared. However, more comparisons between the two sets of accessions are needed for other analyses as in the title (within a collection of the northeast core set and rice landraces), it has been emphasized.

 -In conclusion, the practical use of the results in breeding strategies needs to be discussed.

 - All scientific names of species in references need to be italicized. It is not italicized in some references such as 15., 20, 22, 43.

Author Response

In this research, a multi-locus genome-wide association study was conducted on rice genotypes to identify quantitative trait nucleotides associated with grain-related traits. This study discusses insights from QTNs and candidate genes (CGs) that illuminate rice trait regulation and genetic connections, offering potential targets for future studies.

The analysis of the published data was provided with a sufficient level of scientific novelty. but, there are important flaws in the manuscript listed below:

Response: The authors thank the reviewer for his valuable comments.

Comment 1 -In lines 685 and 686, it is mentioned that "five multi-locus models, namely mrMLM, FASTmrMLM, FASTmrEMMA, pLARmEB, and ISIS EM-BLASSO, were utilized using the R package to pinpoint candidate QTNs". What is the necessity of using five models and what are the advantages of using all these models to pinpoint candidate QTNs?

Response 1: The necessity of employing five models in GWAS allows us to increase the precision and cross-validate the results, thereby increasing the reliability of the findings. Hence genetic association confirmed by several methods is more likely to be positive. Also, the advantages of using all these models increase the power of sensitivity to detect associations, which might be missed otherwise.

 Comment 2 -In lines 712 and 712, two criteria are considered for candidate gene selection. It is unclear if both criteria are necessary for selection or if just one is enough.

Response 2: Thanks for your critical observation. For candidate gene selection, if one satisfies any of the two criteria, it was considered.

 Comment 3- In Figure 1, it is unclear what the correlations are between which characters. Further details and explanation are needed.

Response 3: Thanks for your valuable comment. Figure 1, the correlation between grain-related traits i.e. GL, GW, aroma, and LWR is being studied. Correlation was strong and significant between GL and GW (0.31) and between GL and LWR (0.42). The correlation was negative, strong, and significant between GW and LWR (-0.71). The rest of the correlation values did not turn out to be significant. The same has been incorporated in the main text of the manuscript (lines 138-141)

Comment 4- As two sets of accessions (the northeast core set and the set from other states) are being considered in this study (collection of the northeast core set and rice landraces) and various analyses have been conducted, such as in section 2.5, where the distribution of superior alleles in the northeast core and rice landraces was compared. However, more comparisons between the two sets of accessions are needed for other analyses as in the title (within a collection of the northeast core set and rice landraces), it has been emphasized.

Response 4: Thanks for your suggestion. But the genetic diversity parameters of the northeast core set (Roy Choudhury et al. 2014 [54]) and the rice landrace set (Choudhury et al. 2023 [55]) have already been studied and published earlier, which says about the genetic variation, population structure, unique alleles, and conservation and breeding strategies. 

Comment 5-In conclusion, the practical use of the results in breeding strategies needs to be discussed.

Response 5: Thank you. Practical use of the results in breeding strategies has been included in the manuscript lines 822 onwards.

Comment 6- All scientific names of species in references need to be italicized. It is not italicized in some references such as 15., 20, 22, 43.

Response 6: Done as suggested. All scientific names of species in references have been italicized.

Reviewer 2 Report

Comments and Suggestions for Authors

In this manuscript, the authors conducted a GWAS analysis to identify candidate genomic loci associated with grain quality-related traits in rice populations from India. They successfully identified a set of grain-trait-related candidate genes, offering promising genomic targets for future rice breeding and improvement efforts focused on grain development. The experimental design is rigorous, and the manuscript is structured well. However, the resolution of all figures is extremely low, preventing the reader from obtaining any meaningful information. This significantly hinders the understanding of the manuscript. Specific comments are provided below:

Major concern:

The resolution of the image needs to be substantially improved.

Minor issues:

In the Abstract, the meaning of “CGs” is never mentioned. Does it mean candidate genes?

Line 47. Give the full name of LWR when it was first mentioned. Please check this throughout the main text.

Line 119. Add the statistics of GL.

Display all tables using a three-line grid.

Line 180-181 and line 183-184. Give the detail number rather than “Most”. For lines 180-181,184, it is better to provide a Venn diagram.

Line 191-193. There are only 8 GL-related QTNs, why 12 with LOD scores from 3.04-5.72? A similar question was also in the aroma.

Line 201-203. It is better to remove this part here.

Check the panel ranks in the legend of Figure 6.

Figure 9. Using normalized FPKM for plot drawing.

There is no main text content citing Figure 10.

Line 496-497. On what basis does the author make such an inference?

Line 652. The method involved in SNP genotyping requires further descriptions.

Line 654. The distance between adjacent SNPs should be further checked.

Line 662. Throughout the Materials and Methods section, the authors should provide relevant citations or website links for the software, programs, scripts, or platforms that were utilized in the research.

Line 694-705. It is not clear and there are a lot of redundant descriptions. Also, sections 4.6 and 4.7 are partially repeated.

Line 733-734. How to determine genes with elevated expression levels?

Author Response

In this manuscript, the authors conducted a GWAS analysis to identify candidate genomic loci associated with grain quality-related traits in rice populations from India. They successfully identified a set of grain-trait-related candidate genes, offering promising genomic targets for future rice breeding and improvement efforts focused on grain development. The experimental design is rigorous, and the manuscript is structured well. However, the resolution of all figures is extremely low, preventing the reader from obtaining any meaningful information. This significantly hinders the understanding of the manuscript. Specific comments are provided below:

Response: The authors thank the reviewer for his valuable comments.

 Major concern:

 Comment 1 -The resolution of the image needs to be substantially improved.

Response 1: As suggested the resolution has been made up to 1000 dpi.

 Minor issues:

Comment 2 -In the Abstract, the meaning of “CGs” is never mentioned. Does it mean candidate genes?

Response 2: Thanks. CGs is the short form of candidate genes and have been expanded in the abstract.

Comment 3 -Line 47. Give the full name of LWR when it was first mentioned. Please check this throughout the main text.

Response 3: Thanks for the suggestion. LWR stands for length-width ratio and now it has first been mentioned in the abstract in full form (line 17)

Comment   4 -Line 119. Add the statistics of GL.

Response 4: Thank you. The mean value for GL was missed, hence, the value of GL (5.6) has been added in the manuscript (line 130).

 Comment 5 -Display all tables using a three-line grid.

Response 5: Done as suggested.

Comment 6 -Line 180-181 and line 183-184. Give the detail number rather than “Most”. For lines 180-181,184, it is better to provide a Venn diagram.

Response 6: 19 QTNs overlapped with previous studies (line 203). Venn diagram has been prepared showing the number of overlapping QTNs by various methods and presented in Figure S2.

 Comment 7 -Line 191-193. There are only 8 GL-related QTNs, why 12 with LOD scores from 3.04-5.72? A similar question was also in the aroma.

Response 7: Eight GL-related QTNs were detected, the value 12 depicts the chromosome number (chromosomes 1,2,4,5,8, and 12). Same for aroma.

 Comment 8 -Line 201-203. It is better to remove this part here.

Response 8: Done as suggested. Now lines 222-225 have been removed.

 Comment 9 -Check the panel ranks in the legend of Figure 6.

Response 9: Thank you. The panel ranks have been corrected in Figure 6.

 Comment  10-Figure 9. Using normalized FPKM for plot drawing.

Response 10: Thank you for your valuable suggestion. The legend of Figure 9 has been modified.

 Comment 11 -There is no main text content citing Figure 10.

Response 11: Thank you for your critical observation. Figure 10 has been cited in the main text (line 434)

 Comment 12 -Line 496-497. On what basis does the author make such an inference?

Response 12: Based on the study conducted by Qui et al, 2021 says in rice LD decay of 100 kb and over is best suited to association panels.

 Comment 13-Line 652. The method involved in SNP genotyping requires further descriptions.

Response 13: Done as suggested. The description of the methodology for SNP genotyping has been included under the headings SNP Identification and Array Design, Target Probe Preparation and 50K Rice SNP Array Hybridization, SNP Allele Calling and Data Analysis (please see lines 688 onwards)

Comment 14 -Line 654. The distance between adjacent SNPs should be further checked.

 Response 14: Thank you. The line has been modified accordingly (line 688).

Comment 15 -Line 662. Throughout the Materials and Methods section, the authors should provide relevant citations or website links for the software, programs, scripts, or platforms utilized in the research.

Response 15: As per suggestion. The relevant citations and website have been added in the manuscript in the materials and methods section (please see lines 709 onwards).

 Comment 16 -Line 694-705. It is not clear and there are a lot of redundant descriptions. Also, sections 4.6 and 4.7 are partially repeated.

Response 16: As suggested Line 725 has been reframed. For both candidate gene analysis and functional annotation as well as in-silico gene expression analysis of candidate genes, The Rice Genome Annotation Project Database (RGAP, MSUv7.0, http://rice.uga.edu/) has been used therefore this has been mentioned in both sections.

 Comment 17 -Line 733-734. How to determine genes with elevated expression levels?

Response 17: Based on normalized FPKM values we have determined genes with elevated expression levels. FPKM (fragments per kilobase of exon per million mapped fragments) is a simple expression-level normalization method. The FPKM normalizes read count based on gene length and the relative expression of a transcript, which is proportional to the number of cDNA fragments that originate from it. 

Reviewer 3 Report

Comments and Suggestions for Authors

This is a well written survey of genomic markers associated with quantitative variation for grain related traits.  The premise is simple enough, find the underlying genetic underpinning of important agronomic traits and use this information to find novel haplotypes that can be used in downstream breeding and crop improvement projects.  However, this paper has a powerful method of vetting the ML-GWAS results by using multiple methods and it benefits from low levels of genome wide LD to focus on regions where candidate genes may be uncovered and characterized.  By This paper lays out an approach that drills down into a core collection using a dense 50K SNP chip assay for association analysis.  Like most surveys, the authors report on a whole laundry list of possible candidate loci and they go to great lengths in reporting on the candidate gene functions and expression profiles.  My sense is that this expansive description of potential targets is more aspirational at this point and diverts the reader’s attention from the central elegance in this paper.  Most of the targets have yet to be validated and like many allele mining studies they encumber the paper with scenarios that are potential modes of function each contributing ultimately to the quantitative trait value.            

My advice is to first align the structure of this paper to highlight the process. The conclusion section gives a quick synopsis of the process, and this might benefit from a flowchart diagram for the process that was set up need to come out front and center…without it, the usefulness of this interesting study gets lost in the noise of all the speculative nature of all the various possible targets. I could easily envision a diagram that has a funnel or wedges where the S raw read SNPs are at the top and iterative they are narrowed by bioinformatic filtering, by model congruency by LD blocks.  I would serve as a roadmap for application to other core sets in other crops.

I would suggest keeping the structure roughly congruent with the structure presented in the methods section.  These section headings seem the most reliable in tracking the progress of this study.   Downplay long paragraphs of descriptions of functional descriptions in the main text.  Analyzing the underlying regulatory mechanisms would require additional work and could be a separate paper. I emphasize this because this paper is an excellent example of the practice of allele mining variants to specific LD blocks to candidate genes to functional annotation and ultimately to causative/functional loci.  Of course, this route to the functional significance is one path.  Using the vetted QTNs also have value for training genomic prediction models that can be applied to accessions outside of these core sets. Rice is well developed here both for the diversity of their collections but also for the accuracy of their phenotyping and certainly for the depth of genomic and gene annotation resources.  Other crop may stand to benefit from a clear concise description of the process.

Firstly, the diagrams are uniformly illegible! I struggled to decipher even the most basic descriptive text.  Second many of these figures are supplementary information and really don’t need to be a part of the main manuscript.  Figures 3,6,7,8 and 9 could easily be put into supplementary documentation and the results could be summarized without them.  Figure three can be condensed since they are all complementary, describing the same structure of the collection.  The NJ tree is phenetic, not phylogenetic since no outgroup is included and there is not accommodation for any reticulate pedigree. Figure 5 can be combined and rightly should be brought forward as a major contribution of the study thus far.  These figures, to me, really emphasized the importance and potential of germplasm collections.

Author Response

Please revise the manuscript based on the following comments.

Comment 1: Abstract: Line 13: Include full form of SNPs

Response 1: Thanks for the suggestion and has been modified as suggested.

Materials and Methods: 

Comment 2: Line 640: Please include reference for R package psych. 

Response 2: Thanks, done as suggested (now line 662).

Comment 3: Line 739: Change to "traits of interest"

Response 3: Thanks, change as suggested. (line 765)

Results:

Comment 4: Data in the figures is not clearly visible. Please include high resolution images for all figures.

Response 4: Thanks for the suggestion. High-resolution figures have been uploaded.

Comment 5: If possible, please include a figure depicting the schematic representation of methodology followed in this study.

Response 5: Thank you for your valuable suggestion. A schematic representation of the methodology followed in this study is shown in Figure 11. In the methodology section (line 787).

Comment 6: Line 355: Change to "aerial parts of the plant"

Response 6: Thank you. Done as suggested.

Discussion:

Comment 7: Line 573: Change to "in the current study".

Response 7: Done as suggested.

Comment 8: Please paraphrase the following segments of the manuscript as they are matching with the published material. Lines 78 to 98, 135 to 142, and 642 to 656.

Response 8: Done as suggested.

Comments on the Quality of English Language

Comment 9: Minor editing of English language required. 

Response 9: Thank you for the suggestion. English language editing has been done.

Comment 10: Line 355: Change to "aerial parts of the plant".

Response 10: Done as suggested (now line 377).

Comment 11: Line 573: Change to "in the current study".

Response 11: Done as suggested (now line 800).

Comment 12: Line 739: Change to "traits of interest"

Response 12: Done as suggested (now line 765).

Reviewer 4 Report

Comments and Suggestions for Authors

1). Manuscript ID: Plants-3007668

2). Manuscript Title: Discovering new QTNs and candidate genes associated with rice grain-related traits within a collection of northeast core set and rice landraces

3). Please revise the manuscript based on the following comments.

Abstract: Line 13: Include full form of SNPs

Materials and Methods: 

Line 640: Please include reference for R package psych. 

Line 739: Change to "traits of interest"

Results:

Data in the figures is not clearly visible. Please include high resolution images for all figures.

If possible, please include a figure depicting the schematic representation of methodology followed in this study.

Line 355: Change to "aerial parts of the plant"

Discussion:

Line 573: Change to "in the current study".

5). Comments about iThenticate Report:

 Please paraphrase the following segments of the manuscript as they are matching with the published material.

 Lines 78 to 98, 135 to 142, and 642 to 656.

Comments on the Quality of English Language

Minor editing of English language required. 

Line 355: Change to "aerial parts of the plant".

Line 573: Change to "in the current study".

Line 739: Change to "traits of interest"

Author Response

Comment 1: Abstract: Line 13: Include full form of SNPs

Response 1: Thanks for the suggestion and has been modified as suggested.

Materials and Methods: 

Comment 2: Line 640: Please include reference for R package psych. 

Response 2: Thanks, done as suggested (now line 662).

Comment 3: Line 739: Change to "traits of interest"

Response 3: Thanks, change as suggested. (line 765)

Results:

Comment 4: Data in the figures is not clearly visible. Please include high resolution images for all figures.

Response 4: Thanks for the suggestion. High-resolution figures have been uploaded.

Comment 5: If possible, please include a figure depicting the schematic representation of methodology followed in this study.

Response 5: Thank you for your valuable suggestion. A schematic representation of the methodology followed in this study is shown in Figure 11. In the methodology section (line 787).

Comment 6: Line 355: Change to "aerial parts of the plant"

Response 6: Thank you. Done as suggested.

Discussion:

Comment 7: Line 573: Change to "in the current study".

Response 7: Done as suggested.

Comment 8: Please paraphrase the following segments of the manuscript as they are matching with the published material. Lines 78 to 98, 135 to 142, and 642 to 656.

Response 8: Done as suggested.

Comments on the Quality of English Language

Comment 9: Minor editing of English language required. 

Response 9: Thank you for the suggestion. English language editing has been done.

Comment 1:0 Line 355: Change to "aerial parts of the plant".

Response 10: Done as suggested (now line 377).

Comment 11: Line 573: Change to "in the current study".

Response 11: Done as suggested (now line 800).

Comment 12: Line 739: Change to "traits of interest"

Response 12: Done as suggested (now line 765).

Reviewer 5 Report

Comments and Suggestions for Authors

The manuscript “Discovering new QTNs and candidate genes associated with rice grain-related traits within a collection of northeast core set and rice landraces” reports several analyzes aimed at the selection and characterization of candidate genes related to rice grain-related traits. It is, therefore, certainly of interest to the researchers involved in similar studies.

Since the figures represent an integral part of the manuscript and they are important for an adequate and rapid understanding of the results presented, I believe it is absolutely necessary that the quality of all the figures reported be improved as they are all partially or completely illegible.

Furthermore, it seems to me that figure 10, reported in the manuscript, is not described in the text.

Author Response

The manuscript “Discovering new QTNs and candidate genes associated with rice grain-related traits within a collection of northeast core set and rice landraces” reports several analyzes aimed at the selection and characterization of candidate genes related to rice grain-related traits. It is, therefore, certainly of interest to the researchers involved in similar studies.

Comment 1: Since the figures represent an integral part of the manuscript and they are important for an adequate and rapid understanding of the results presented, I believe it is absolutely necessary that the quality of all the figures reported be improved as they are all partially or completely illegible.

Furthermore, it seems to me that figure 10, reported in the manuscript, is not described in the text.

Response 1: Thank you for your critical observation. Figure 10 has been cited in the main text (line 434) and high-resolution figures have now been uploaded as suggested.

Round 2

Reviewer 2 Report

Comments and Suggestions for Authors

I cannot see Figures 1, 2, 3, 5, and 8 in the main text. Also, some issues I asked for should be further corrected.

Comment 13. The description and logic presented in section 4.3 were somewhat perplexing to me and could be improved by providing a more step-by-step, clear, and detailed explanation. Additionally, vague phrases like "software like PLINK" should be avoided. The author should explicitly state which software was utilized and provide a truthful and thorough description of its purpose and functionality. All M&M should be performed like this and the authors should pay more attention to this!

Comment 14. The rice genome size is 400Mb, with a total of 50,000 SNPs, and the average distance between SNPs is approximately 8Kb.

Comment 17. What I want to ask is how to detect up-regulated genes (differentially expressed genes) rather than quantify expression level. Related software should be provided.

Author Response

Comments and Suggestions for Authors

I cannot see Figures 1, 2, 3, 5, and 8 in the main text. Also, some issues I asked for should be further corrected.

Response: Figures 1,2,3,5 and 8 have been modified. Also, issues with the rest of the figures have been resolved to make the text legible.

Comment 13. The description and logic presented in section 4.3 were somewhat perplexing to me and could be improved by providing a more step-by-step, clear, and detailed explanation. Additionally, vague phrases like "software like PLINK" should be avoided. The author should explicitly state which software was utilized and provide a truthful and thorough description of its purpose and functionality. All M&M should be performed like this and the authors should pay more attention to this!

Response 13: The material method section has been revised, with the suggestions. Some sentences have been deleted (lines 762 onwards) and related software and databases have been cited.

Comment 14. The rice genome size is 400Mb, with a total of 50,000 SNPs, and the average distance between SNPs is approximately 8Kb.

Response 14: The average distance between adjacent SNPs is 0.745 kbp as stated by Singh et al. (2015) [99]

Comment 17. What I want to ask is how to detect up-regulated genes (differentially expressed genes) rather than quantify expression level. Related software should be provided.

Response 17: Thank you for your question

Regarding the detection of up-regulated genes. We used the Rice Genome Annotation Project (RGAP) database to obtain FPKM expression values (http://rice.uga.edu/expression.shtml) for various tissues, including leaves (20 days), post-emergence inflorescence, pre-emergence inflorescence, anther, pistil, seed (5 days after pollination (DAP)), embryo (25 DAP), endosperm (25 DAP), seed (10 DAP), and shoots. All 60 candidate genes (CGs) expression values were visualized using a heatmap (Figure 9, Table S6). We have compared the FPKM values among the mentioned tissues.

We downloaded the complete expression data table for all rice genes from the Rice Expression Matrix File, available in Excel, and zipped tab-delimited text file formats in the RGAP database. The expression data were derived from the NCBI Sequence Read Archive (SRA). Sequence reads were mapped to the version 7 pseudomolecules using Tophat (Trapnell et al.,2009), and RNA-Seq libraries' expression abundances were calculated with Cufflinks (Trapnell et al., 2010). This comprehensive approach allowed us to detect and analyze candidate genes across various mentioned tissues and developmental stages in rice.